# NBEAL2 deficiency in humans leads to low CTLA-4 expression in activated conventional T cells

Loss of NBEAL2 function leads to grey platelet syndrome (GPS), a bleeding disorder characterized by macro-thrombocytopenia and α-granule-deficient platelets. A proportion of patients with GPS develop autoimmunity through an unknown mechanism, which might be related to the proteins NBEAL2 interacts with, specifically in immune cells. Here we show a comprehensive interactome of NBEAL2 in primary T cells, based on mass spectrometry identification of altogether 74 protein association partners. These include LRBA, a member of the same BEACH domain family as NBEAL2, recessive mutations of which cause autoimmunity and lymphocytic infiltration through defective CTLA-4 trafficking. Investigating the potential association between NBEAL2 and CTLA-4 signalling suggested by the mass spectrometry results, we confirm by co-immunoprecipitation that CTLA-4 and NBEAL2 interact with each other. Interestingly, NBEAL2 deficiency leads to low CTLA-4 expression in patient-derived effector T cells, while their regulatory T cells appear unaffected. Knocking-down NBEAL2 in healthy primary T cells recapitulates the low CTLA-4 expression observed in the T cells of GPS patients. Our results thus show that NBEAL2 is involved in the regulation of CTLA-4 expression in conventional T cells and provide a rationale for considering CTLA-4-immunoglobulin therapy in patients with GPS and autoimmune disease.

Recessive mutations in the *NBEAL2* gene lead to grey platelet syndrome (GPS), characterized by macrothrombocytopenia, α-granule-deficient platelets, bleeding disorders, and (in some cases) splenomegaly and progression to myelofibrosis[1–3].

The clinical phenotype of *Nbeal2* knock-out mice recapitulates human GPS[3,4]. Moreover, several immune cell anomalies have been observed in *Nbeal2*[−/−] mice. The granule count in neutrophils was abnormally low, and natural killer (NK) cell degranulation was impaired – leading to greater susceptibility to bacterial and viral infections (with *Staphylococcus aureus*, and murine cytomegalovirus, respectively)[5]. After the induction of systemic inflammation with lipopolysaccharide (LPS), the *Nbeal2*[−/−] mice's circulating monocyte and neutrophil counts were found to be abnormally low[6]. The observations in mouse models therefore suggest that *NBEAL2* has a role in immune cell function.

Initially, GPS was described as a platelet disorder. However, some NBEAL2-deficient patients present clinical features of autoimmune lymphoproliferative syndrome (ALPS)[7] – suggesting a role for NBEAL2 in immune homeostasis and tolerance. More recently, characterization of a broad international cohort of patients with GPS revealed immune system abnormalities (autoimmune disease, autoantibodies, leukopenia, etc.) in 81% of the individuals[8]. The consequences of recessive *NBEAL2* mutations on neutrophils' biological functions have also been described[5,9]. Neutrophils from patients with GPS exhibit defective neutrophil extracellular trap (NET) formation (also referred to as NETosis) and contain an abnormally low number of specific granules (due to premature release)[9]. The early release of granule contents is also observed in platelets, where the α-granules cargoes are not retained in the platelets[10]. Although NBEAL2's role in granule retention

✉ e-mail: frederic.rieux-laucat@inserm.fr

has been well described, the mechanism leading to the development of autoimmune manifestations in patients with GPS has not.

NBEAL2 is a member of the Beige and Chediak (BEACH) domain-containing protein (BDCP) family. It is composed of several domains: a BEACH domain, a pleckstrin homology domain, a concanavalin-A-like lectin domain, an armadillo-like domain, and WD40 repeats[11,12]. The BDCP family comprises nine members, all of which contain a conserved BEACH domain and have been characterized as scaffold proteins involved in vesicular trafficking[13]. LPS-responsive beige-like anchor protein (LRBA) is a BEACH family member with structural similarity to NBEAL2[13]. LRBA is involved in regulating the endocytosis of ligand-activated receptors. In particular, it prevents the lysosomal degradation of CTLA-4 by physically interacting with the latter in RAB11+ endosomes[14]. CTLA-4 is an inhibitory immune checkpoint that competes with CD28 (a co-stimulatory immune checkpoint) for the ligands CD80/CD86 and thereby has a crucial role in regulating T-cell-mediated immune responses[15]. CTLA-4 is constitutively expressed at the surface of regulatory T cells (Treg) and (following activation) conventional T cells[16]. In humans, heterozygous mutations of *CTLA-4* are associated with autoimmunity (diabetes, enteropathies, and cytopenia), lymphocyte infiltrations, and progressive B lymphopenia resulting in lung infections[17,18]. In humans, primary LRBA deficiency leads to a secondary deficiency in CTLA-4 expression and thus is associated with immune dysregulation and autoimmune manifestations similar to those observed in patients with a primary CTLA-4 deficiency[14,19].

To extend our knowledge on autoimmunity and immune abnormalities in patients with GPS, we investigate NBEAL2's role in the immune system and particularly in T cells. Taken as a whole, our data reveal a new function for NBEAL2: the regulation of the CTLA-4 expression in proliferating conventional T cells, but not in regulatory T cells. Our findings thereby provide an explanation for the immune dysregulation observed in patients with GPS and open a possible CTLA4-Ig therapeutic option to treat their autoimmune manifestations.

## Results

### Autoimmune manifestations in patients with NBEAL2 deficiency

We studied 11 patients with GPS (out of a cohort of 13) from nine unrelated families (Supplementary Fig. 1A); all carried recessive *NBEAL2* mutations (Table 1), which variously affected several domains (Fig. 1a). Most of the patients have previously been reported in other studies, as stated in Supplementary table 1. Very low levels of NBEAL2 protein (P3, P4.1) or the absence of NBEAL2 protein were observed in lysates prepared from the patients' activated T cells, thus confirming the harmful impact of the recessive *NBEAL2* mutations (Fig. 1b).

All patients exhibited a bleeding tendency (epistaxis and bruising) and grey platelets lacking α-granules on blood smears. Splenomegaly was noted in four patients (Table 1). One patient (P3) had several episodes of Epstein-Barr virus (EBV) reactivation (>3 log), without macrophage activation syndrome, that were finally controlled after four courses of anti-CD20 therapy. After B cell recovery, a mild hypogammaglobulinemia (6 mg/dl) persisted without related infection. One patient (P4.2) underwent allogenic hematopoietic stem cell transplantation due to bone marrow fibrosis[20]. Bone marrow biopsies were performed in five patients and myelofibrosis was diagnosed in 4 (Table 1).

As described previously[7], abnormal laboratory results reminiscent of those observed in patients with ALPS (such as elevated plasma levels of vitamin B12 and Fas ligand (FasL)) were observed in patients with GPS (Supplementary Fig. 1B, C). Five patients had autoimmune manifestations, including Evans syndrome, chill blain lupus, autoimmune thyroiditis, and/or antiplatelet autoantibodies (Table 1). NBEAL2 deficiency's frequent association with features of autoimmunity and immunological abnormalities[8] suggests that the protein has a role in immune homeostasis and tolerance to self.

**Table 1 | Genetics, main immunological abnormalities, and auto-immune manifestations in the grey platelet syndrome patients cohort**

| Patient ID | Gender | Age range at sampling | Affected Allele | Mutation 1 | Mutation 2 | Affected protein domains | Immunological abnormalities and autoimmune manifestations | Splenomegaly | Myelofibrosis |
|---|---|---|---|---|---|---|---|---|---|
| P1 | M | 18-25 yo | Homozygous | p.R1839C | p.R1839C | – | Evans syndrome, neutropenia, adenopathy | – | – |
| P2 | M | 10-18 yo | Homozygous | p.L1501X | p.L1501X | – | Chill blain lupus | – | – |
| P3[a] | F | 18-25 yo | Homozygous | p.I682F | p.I682F | Con A | Multiple episodes of EBV reactivation | Splenomegaly | – |
| P4.II-1# | F | 35-45 yo | Compound | p.E643V | p.P2100L | Con A & BEACH | Recurrent infections | – | Myelofibrosis |
| P4.II-2# | M | 35-45 yo | Compound | p.E643V | p.P2100L | Con A & BEACH | – | – | Myelofibrosis leading to BMT |
| P4.II-3# | F | 35-45 yo | Compound | p.E643V | p.P2100L | Con A & BEACH | Auto-immune thyroiditis, RA, vitiligo | – | Myelofibrosis |
| P5.II-1 | F | 55-65 yo | Homozygous | p.R1631Gfs*3 | p.R1631Gfs*3 | – | Auto-immune thyroiditis | Splenomegaly | – |
| P5.II-2 | M | 55-65 yo | Homozygous | p.R1631Gfs*3 | p.R1631Gfs*3 | – | Moderate CD4 + T and B lymphopenia, platelets autoantibodies | Splenomegaly | – |
| P6# | M | 45-55 yo | Compound | p.Y1020H | c.2650-1 G > A | – | – | – | Myelofibrosis |
| P7# | M | 25-35 yo | Compound | p.S2269L | p.G2553E | BEACH & WD40 | NK lymphopenia | – | – |
| P8 | F | 45-55 yo | Compound | p.C2190Xfs*23 | p.L2646P | BEACH & WD40 | T and NK lymphopenia, alopecia | Splenomegaly | – |
| P9.II-1* | M | 18-25 yo | Homozygous | p.T2487fs*16 | p.T2487fs*16 | WD40 repeats | – | – | – |
| P9.II-2* | F | 10-18 yo | Homozygous | p.T2487fs*16 | p.T2487fs*16 | WD40 repeats | – | – | – |

The variants not previously described in the literature are mentioned by a star (*). All variants are detailed in Supplementary Table 1. Bone marrow biopsies were performed for patients mentioned by the symbol (#).
*GPS* grey platelet syndrome, *M* male, *F* female, *yo* years old, *BMT* bone marrow transplant.
[a]Patient P3 is described for the first time, however the same variant has previously been described (see Supplementary Table 1).

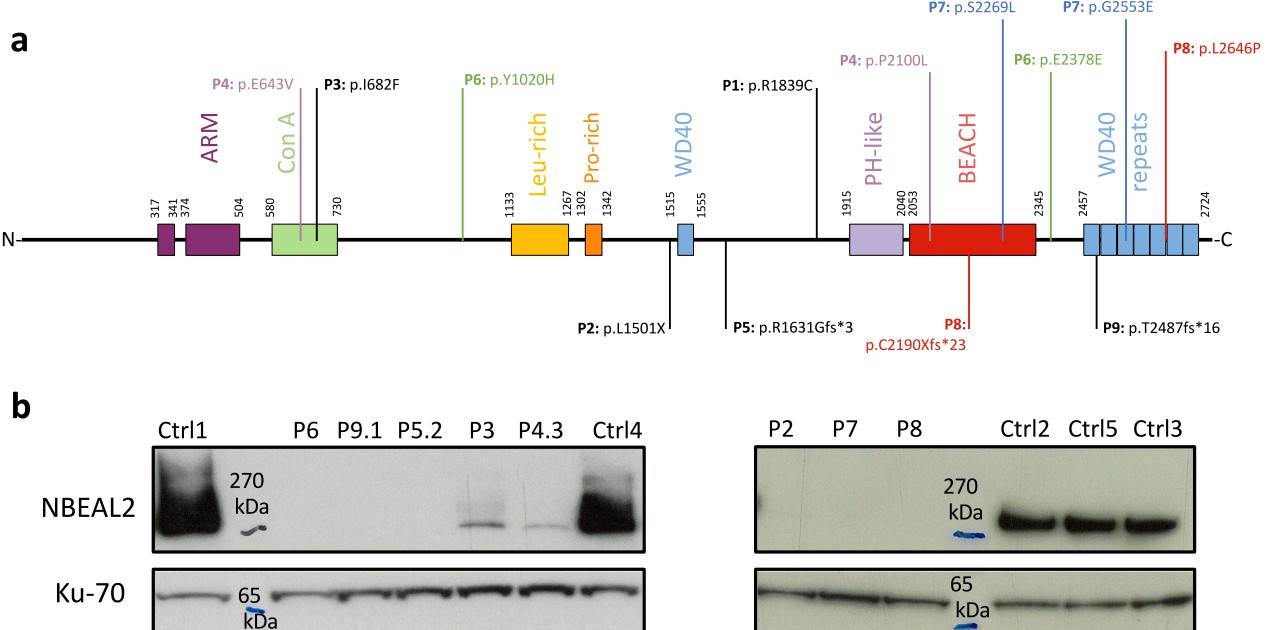

**Fig. 1 | Grey platelet syndrome cohort: mutations characteristics. a** Positions of the *NBEAL2* mutations at the protein level. Compound heterozygous mutations are annotated with the same color. Homozygous mutations are annotated in black. **b** NBEAL2 immunoblotting on activated T cells lysates from patients and healthy controls (Ctrl), performed once for each sample. The protein ku-70 is used as a loading control. Source data are provided as a Source Data file.

## Immunophenotyping of patients with GPS

T-cell lymphopenia (in patients P6 and P8), B cell lymphopenia (P6) or NK cell lymphopenia (P7, P8) was found in three patients (Table 1). To further analyze the immune phenotypes of patients with GPS, standardized immunostaining by cytometry by time of flight (CyTOF) were performed on fresh blood samples. The results for 37 markers enabled us to cluster the immune cells into 34 identified subsets (Fig. 2A and supplementary Fig. 2A). Seven patients with recessive *NBEAL2* mutations (P1, P2, P5.2, P6, P7, P9.1, and P9.2) and 11 age- and sex-matched healthy controls were analyzed. A visual comparison of the uniform manifold approximation and projection (UMAP) results suggested an overall difference in marker expression by the patients' neutrophils, together with an abnormally high γδ T-cell count (Fig. 2a, red arrows). A cluster bias analysis revealed several abnormalities, including significantly elevated proportions of naive B cells, Tregs, and central memory and effector memory CD8+ T cells in the patients (Fig. 2b). A significant elevation of the proportion of T follicular helper (TFH) CD4+ memory T cells was also observed (Fig. 2b and Supplementary Fig. 2b). Monocytes, neutrophils, conventional dendritic cells (cDC), double-negative T cells, and CD8+ T cells expressed abnormally low levels of CD45RA (Fig. 2C). Furthermore, FAS was overexpressed by eosinophils, B cells, and cDCs. HLA-DR and CD66b expression levels on eosinophils and neutrophils were abnormally high, which suggested activation and premature degranulation (Fig. 2c and Supplementary Fig. 2c, d). It is noteworthy that elevated CD66b expression by neutrophils in patients with GPS has been reported previously[9]. The overexpression of PD1 by mucosal-associated invariant T (MAIT) cells and CD4+ T cells also suggests that these subsets are activated in patients with GPS. Lastly, TIM3, HLA-DR and CD11c expression levels were elevated in NK cells, whereas CXCR3 expression was lower; again, this suggests activation[21–23]. Taken as a whole, these results highlighted the activation of several leukocyte subsets in patients with GPS.

## NBEAL2's partners in activated T cells

To characterize NBEAL2's functionality, we analyzed the interactome by immunoprecipitating the protein in lysates of activated T cells. To ensure that the analysis was specific, two healthy subjects were compared with two patients with GPS (P6 and P7). Immunoblotting of NBEAL2 in the input fractions showed that the protein was barely expressed in P7 and could not be detected in P6 (Fig. 3a). Following NBEAL2 pull-down, PageBlue staining confirmed the enrichment of NBEAL2 in controls but not in patients with GPS (Fig. 3a). The IP fractions were then analyzed using mass spectrometry and 3527 proteins were identified. With our protein selection strategy (Supplementary Fig. 3) and a comparison of control versus patient samples, the partner list was curated to give 74 partners for NBEAL2 with high intensities and high control/patient ratios (Fig. 3b). To validate this selection and rule out possible bias due to differential expression, we measured protein expression levels in the input fractions (using mass spectrometry) and found no difference between control and patient samples. Finally, to confirm the potency and stringency of this selection, we compared our initial dataset (3527 proteins before selection) to a list of 133 published potential partners of NBEAL2[10,24,25] (Supplementary Table 2). Eighty of the potential partners previously described were in our dataset. Among these common proteins, SEC22B, Vac14, and SEC16A were present. From our data, we calculated the ratio of intensities between controls and patients (as explained in Fig. S3) and sorted the 80 common proteins by controls/P6 ratio (as in Fig. 3B). Vac14 and SEC16A, known specific partners of NBEAL2, ranked in the top 6. This comparison highlights the potency of our analyses to select specific partners. Of note, although SEC22B, Vac14, and SEC16A are specific partners of NBEAL2, they were not retained in our list of 74 candidate partners because of the stringency of our selection criteria (Supplementary Fig. 3).

A network analysis of the 74 selected proteins was performed using the STRING database[26]; cytoskeleton binding proteins and proteins involved in immune responses, protein transport, and GTPase binding were found (Fig. 3c). In an enrichment analysis using the EnrichR website[27], CCDC88B and DOCK8 gave some of the strongest associations with NBEAL2 (Fig. 3b). Interestingly, both of these proteins are known to be involved in the transport of lytic granules[28].

To measure the expression of NBEAL2 partners in other immune cell types, we used the Jensen Tissues library (a weekly-updated database that integrates evidence on tissue expression from manually

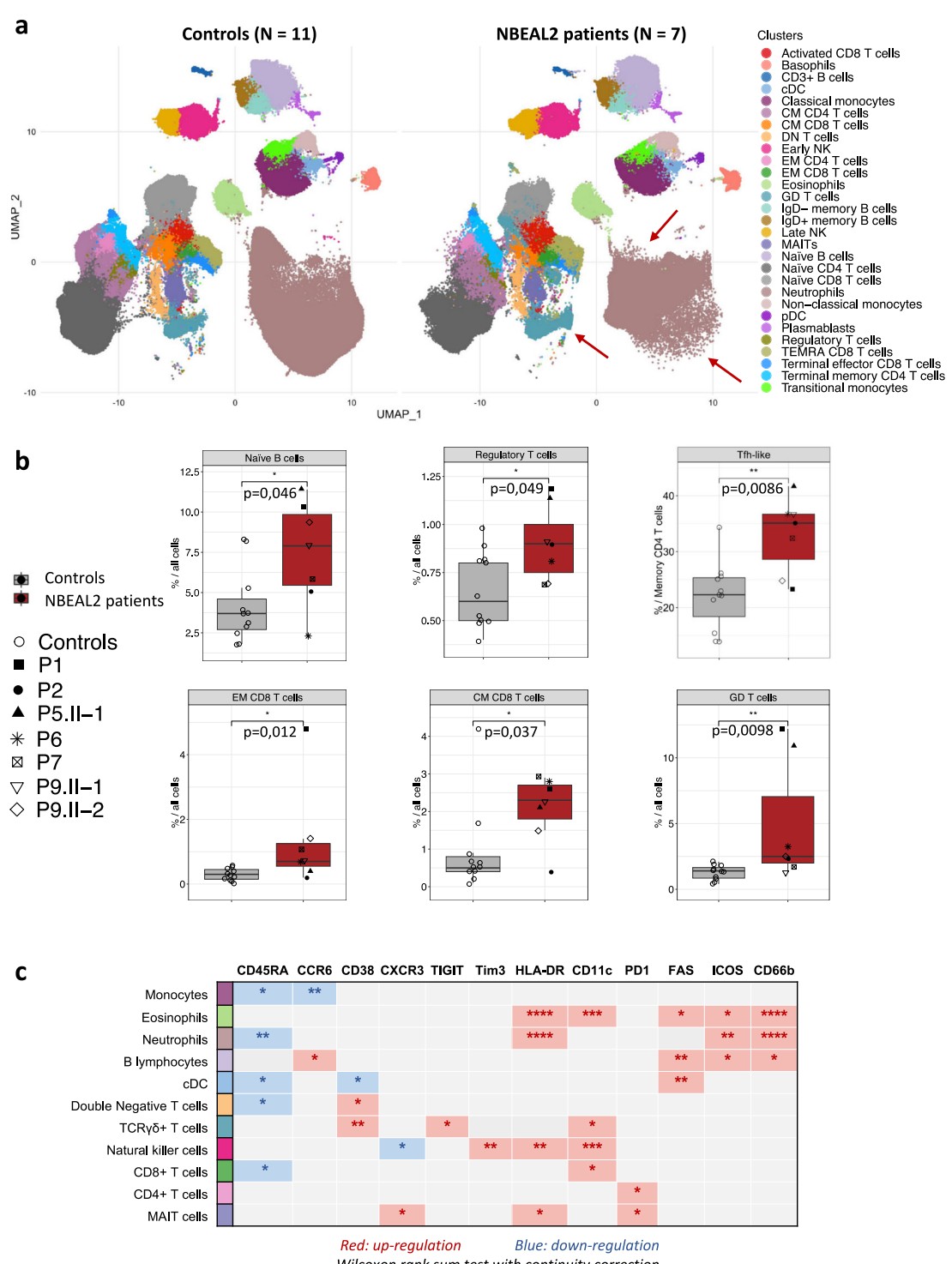

**Fig. 2 | Immunophenotyping of patients with GPS. a** Uniform manifold approximation and projection (UMAP) of immune cell subsets analyzed by CyTOF on whole blood samples from age-matched healthy controls (N = 11) and NBEAL2 patients (N = 7). Each cluster is color coded. The red arrows indicate changes in the cluster's shapes of the patients group. The markers used to identify these cell clusters are shown in the heatmap of supplementary figure 2 A. cDC classical dendritic cell, CM central memory, EM effector memory, pDC plasmacytoid dendritic cell, TEMRA terminally differentiated effector memory re-expressing CD45RA, NK natural killer, MAIT mucosal associated invariant T cell. **b** Statistically significant cluster biases observed in the CyTOF immune phenotype from healthy controls and NBEAL2 patients. Results shown correspond to 11 controls and 7 NBEAL2 patients. The line at inside the box is the median value (50th percentile). Minima and maxima of the boxes correspond to 25th and 75th percentile. Whiskers marks the 10th and 90th percentile. Two-tailed *p*-values were determined with a Wilcoxon rank-sum test with continuity correction. *p*-value < 0.05; **p*-value < 0.01; ****p*-value < 0.005; *****p*-value < 0.0001. TFH-like T follicular helper cells, EM effector memory, CM central memory, GD T cells: γδ T cells. Source data are provided as a Source Data file. **c** Table summarizing the significant overexpression (red) or downregulation (blue) of biological markers in the different immune cell subsets from GPS patients versus healthy controls. Two-tailed *p*-values were determined with a Wilcoxon rank sum test with continuity correction. *p*-value < 0.05; **p*-value < 0.01; ****p*-value < 0.005; *****p*-value < 0.0001.

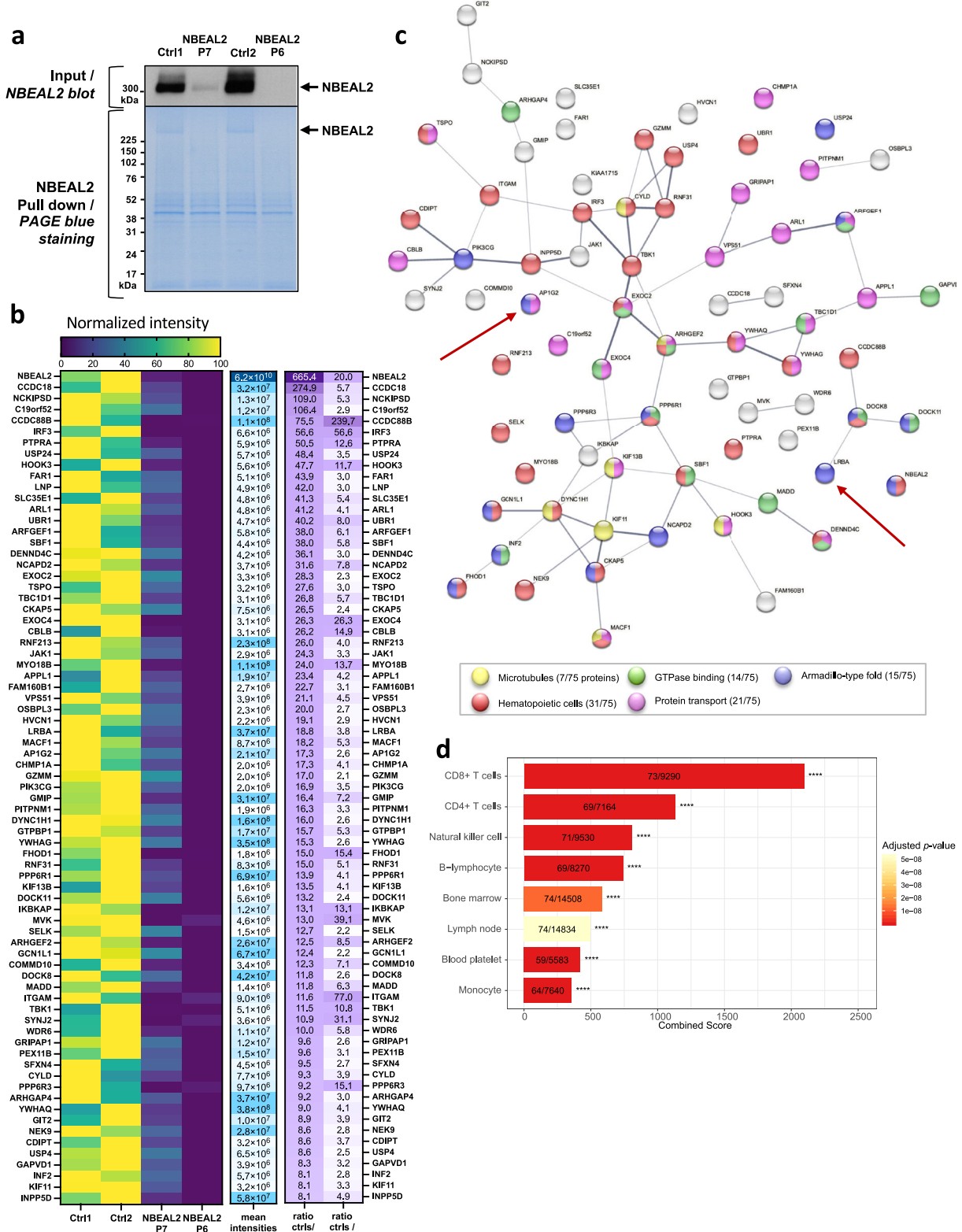

curated literature, proteomics, and transcriptomics screens)[29]. We found that almost all of the 74 proteins were also expressed in other cell types, i.e. NK cells, B lymphocytes, platelets, and monocytes (Fig. 3d). Thus, the interactions between NBEAL2 and its partners might not be restricted to activated T cells. Finally, we performed a bulk RNAseq analysis on activated T cells to verify whether the proteins absent in GPS patients are the consequence of transcriptomic

regulation or a consequence of the absence of NBEAL2. For the 21 proteins found absent in patients' proteome, we observed identical expression at the RNA level between the patient cells and the control cells (Supplementary Table 4). They could therefore be proteins whose expression is disturbed as the consequence of the NBEAL2 deficiency either by a defect in intracellular sorting or by another mechanism leading to their increased destruction.

**Fig. 3 | NBEAL2's partners in activated T cells. a** Immunoblotting of NBEAL2 in T cells lysate from healthy subject (Ctrl 1 and 2) and NBEAL2 patients (P6, P7). Immunoblot of the input fractions before NBEAL2 immunoprecipitation in T cells (top) and Page Blue staining after NBEAL2 pull down (bottom). Experiment performed once. **b** Heatmaps of the selected NBEAL2 partners after mass spectrometry analyses. The left heatmap shows normalized intensities of the 74 proteins selected after mass spectrometry analysis of the NBEAL2 interactome. The middle heatmap represents the mean intensities detected for the two controls (by mass spectrometry). The heatmap on the right shows the intensities ratio of the mean control intensity over each NBEAL2 patient. The proteins are ranked by intensities ratio of controls over NBEAL2 patient P6. **c** STRING network analysis of the 74 proteins and node coloring per function. The network edges indicate both functional and physical protein associations: proteins jointly contribute to a shared function. However, the network edges do not necessarily mean proteins are physically binding to each other. The line thickness indicates the strength of data support. **d** Analysis of the 74 proteins partners of NBEAL2 in enrichR using the Jensen Tissue library. The most represented cells in which the proteins partners are expressed is shown. In the barplots the number of proteins versus the number of proteins defining the cell type is indicated. Two-sided $p$-value is obtained with a Fischer's exact test and adjusted with a Benjamini–Hochberg correction. The combined score is a combination of the $p$-value and z-score calculated by multiplying the two scores. Source data are provided as a Source Data file.

A striking finding was that LRBA and AP1G2 (γ2 subunit of AP-1 complex) were NBEAL2 partners; the two proteins are thought to be involved in CTLA-4 trafficking (Fig. 3b, c). In view of NBEAL2-LRBA interaction and LRBA's known function in CTLA-4 recycling, we investigated NBEAL2's role in CTLA-4 expression further.

### NBEAL2's role in the regulation of CTLA-4 expression

As Tregs constitutively express CTLA-4, we first investigated the latter protein's expression in Tregs from patients with GPS. The mean fluorescence intensity (MFI) of CTLA-4 and the percentage of CTLA-4$^+$ Tregs were similar in patients and healthy controls (Figs. 4a–c and S4). CTLA-4 MFI on other T-cell subsets from non-activated PBMC were also similar between patients and controls (Supplementary Fig. 4d). In contrast, the CTLA-4 MFI and the percentage of T cells expressing CTLA-4 following TCR-activation was significantly lower among patients with GPS (Fig. 4d, e and Supplementary Fig. 4g) than in healthy donors. The cut-off for determining CTLA-4 positive T cells after activation is shown in Fig. 4d and Supplementary Fig. 5a, c.

The level of CD25 expression was similar in all experiments, which ruled out defective stimulation of the patients' T cells (Supplementary Fig. 5b, d–f). These observations suggest that NBEAL2 has a role in the regulation of CTLA-4 expression following T-cell activation. Moreover, we measured the expression of NBEAL2 and LRBA proteins in Tregs and conventional CD4$^+$CD25$^-$ T cells (Tconv) from 3 healthy donors (Supplementary Figs. 6 and 7). Total proteins were extracted from sorted Tregs and Tconv as well as from in vitro activated Tconv. NBEAL2 is not, or faintly, detected in Tregs, but it is well expressed in Tconv (Supplementary Fig. 6d, e). Importantly, NBEAL2 expression increased in activated Tconv. For LRBA, we observed a mirror expression profile compared to NBEAL2. LRBA expression is high in Tregs and Tconv, but its expression is weaker in in vitro activated Tconv. These results show that NBEAL2 is not, or very weakly, expressed in Tregs. This observation is, therefore, consistent with the normal expression of CTLA-4 in the Tregs of NBEAL2-deficient GPS patients.

To confirm the effect of NBEAL2 deficiency on CTLA-4 expression, we used CRISPR-Cas9 technology to knock down *NBEAL2* in activated primary T cells. In parallel, LRBA was knocked down as an internal control. Six mRNA guides were designed for NBEAL2, and 5 were designed for LRBA (Fig. 4f, g). Immunoblotting of NBEAL2 and LRBA were provided information on the consequences of the knockdown on the protein level (Fig. 4h, i). All six NBEAL2 guides and three of the five LRBA guides were effective for knocking down expression of their respective target protein (Fig. 4h, i). The remaining cells were stained for CTLA-4. The CTLA-4 MFI in T cells mirrored closely the NBEAL2 knockdown (Fig. 4j). In control cells with LRBA knockdown, CTLA-4 expression was low in the conditions previously found to be associated with the absence of LRBA protein expression (Fig. 4j). These results confirmed the observations obtained in activated T cells from patients with GPS (Fig. 4d, e) and again suggested that NBEAL2 has a role in the regulation of CTLA-4 expression.

We next used a CTLA-4 pull-down approach to determine whether NBEAL2 and CTLA-4 interacted in primary T cells from healthy donors.

Non-stimulated T cells were used as a negative control for comparison with activated T cells (since CTLA-4 is expressed after activation)[30,31]. In CD3/CD28-activated T cells, CTLA-4 co-immunoprecipitated with LRBA (as expected)[14], and with NBEAL2 (Fig. 4k, left panel). The specificity of these interactions was confirmed, since neither NBEAL2 nor LRBA was pulled down in non-stimulated T cells not expressing CTLA-4. These results highlight the interaction between CTLA-4 and NBEAL2.

### Single-cell RNA sequencing of NBEAL2- and LRBA-deficient cells

Single-cell RNA sequencing experiments were carried out on peripheral blood mononuclear cells (PBMC). The rationale for this single-cell analysis was to compare dysregulated pathways in a known CTLA-4 deficiency (i.e., LRBA deficiency) to the dysregulated pathways in GPS immune cells. Data from seven age- and sex-matched healthy controls were compared with data from five NBEAL2-deficient patients (P2, P4.II-1, P5.II-1, P7, P8) and five LRBA-deficient patients. Thirty-five clusters were observed and annotated (Supplementary Figs. S8–S10). Quality controls were performed (Supplementary Table 3 and Supplementary Fig. 11). The expression level of *CTLA-4*, *LRBA*, and *NBEAL2* were determined in control T-cell subsets, which were ranked in terms of *NBEAL2* expression (Supplementary Fig. 9b). The highest *NBEAL2* expression was found in proliferating T cells. CTLA-4$^+$ Tregs expressed *LRBA* but had a lower *NBEAL2* expression level. These results are correlated with the findings obtained in NBEAL2-deficient Tregs, which expressed CTLA-4 to much the same extent as Tregs from healthy controls (Fig. 4a). Altogether, our data suggest that NBEAL2 is not essential for CTLA-4 expression in Tregs but is required in activated effector T cells.

Considering NBEAL2's potential role in regulating CTLA-4 expression in activated effector T cells, we analyzed differentially expressed genes (DEG) in control, NBEAL2-deficient and LRBA-deficient CD4$^+$ and CD8$^+$ T cells. The pathways found to be dysregulated are shown in Fig. 5a (for CD4$^+$ T cells) and 5b (for CD8$^+$ T cells). These pathways were similar in LRBA- and NBEAL2-deficient CD4$^+$ T cells and CD8$^+$ T cells, suggesting that the transcriptional dysregulations were similar. Pathways related to cytoskeleton and lymphocyte activation (especially "TCR signaling" and "CD28 signaling in T helper cells") were found to be dysregulated. Of note, Sims et al.[8] have already shown an overrepresentation of immune dysregulated functions after RNA bulk sequencing of CD4$^+$ T cells. Lastly, we performed a DEG analysis of NBEAL2- versus LRBA-deficient cells, the results of which highlighted an upregulation of gene expression in the IL6-STAT3 pathway (Supplementary Fig. 12). Hence, we plotted the Hallmark IL-6/STAT3/JAK signature from MSigDB (Fig. 5C)[32,33] in control, NBEAL2-deficient and LRBA-deficient groups. The results showed that activation of the STAT3/IL-6/JAK pathway was stronger in LRBA-deficient effector and memory T cells than in their NBEAL2-deficient counterparts.

## Discussion

NBEAL2 was initially described as a key factor for platelet α-granule biogenesis - a function discovered in patients with GPS carrying

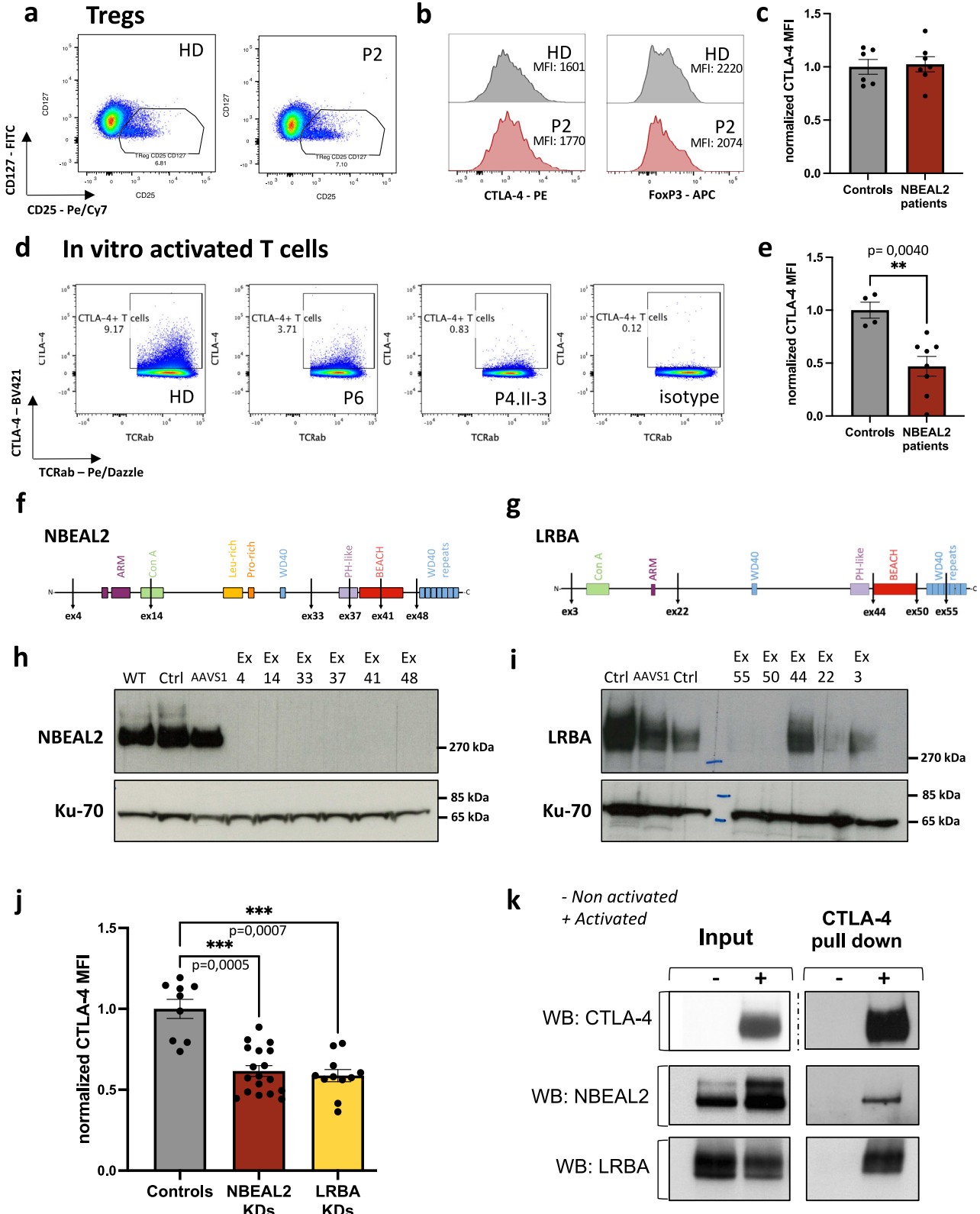

biallelic *NBEAL2* mutations[1,2,11]. GPS is a rare, autosomal, recessive, monogenic disease mainly characterized by mild/moderate bleeding disorder, thrombocytopenia, and a lack of platelet α-granules. However, reports of autoimmune manifestations and immune dysregulation have emerged over time[7,8]. These observations suggested that NBEAL2's role extends beyond platelet biogenesis and are consistent with a broad expression profile – particularly in immune cells[8,9,34,35]. By

extensively immunophenotyping blood cells from patients with GPS, we evaluated the overall impact of NBEAL2 deficiency on innate and adaptive immune cell subsets. We showed that the patients' cells exhibited an activated phenotype, with abnormally high expression of HLA-DR and FAS and abnormally low expression of CD45RA. The slightly elevated naïve B cell, Treg, memory CD8+ T-cell and T follicular helper cell counts have not previously been described in patients with

**Fig. 4 | NBEAL2 is involved in CTLA-4 expression after TCR activation.**
**a** Representative gating strategy for one healthy donor (HD) and one NBEAL2-deficient patient on CD4 + T cells. Tregs were defined as CD4$^+$CD25$^+$CD127$_{low}$ T cells. **b** Representative histograms of CTLA-4 and FOXP3 expression on CD25$^+$CD127$_{low}$ Treg for one healthy donor (HD) and one NBEAL2-deficient patient. **c** Normalized CTLA-4 mean of fluorescence intensity (MFI) in CD25$^+$CD127$_{low}$ Treg from controls ($N = 6$) and NBEAL2-deficient patients ($N = 7$). Mean ± SEM is represented. **d** Gating strategy to define CTLA-4 positive T cells after activation. The gating was tuned using a control isotype antibody. **e** Normalized CTLA-4 mean of fluorescence intensity (MFI) of activated T cells from controls ($N = 4$) and NBEAL2-deficient patients ($N = 8$). Mean ± SEM is represented. Two-tailed $p$-value is determined with a non-parametric Mann–Whitney test. **f, g** Diagram of the primary structure of NBEAL2 (**f**) and LRBA (**g**) proteins. Positions targeted by the CRISPR guide RNA used are indicated. **h, i** NBEAL2 (**h**) and LRBA (**i**) immunoblotting after CRISPR-Cas9 experiment on activated T cells. WT: refers to cells that underwent nucleofection without CRISPR/Cas9 in the solution. Ctrl: refers to cells nucleofected with a scramble guide and the Cas9. AAVS1: refers to cells transfected with a the Cas9 and a guide targeting an intronic position of the genome. Other cell lysates correspond to NBEAL2-CRISPR/Cas9 treatment (**f**) and LRBA-CRISPR/Cas9 treatment (**g**). Ku-70 is used as a loading control. **j** Normalized CTLA-4 MFI after CRISPR-Cas9 knockdown (KD) experiment for each mRNA guide (6 for NBEAL2; 5 for LRBA) used on three independent healthy donors activated T cells, over 2 independent experiments. Mean ± SEM is represented. Controls correspond to the three conditions Ctrl, WT, and AAVS1 described in **h** and **i** for each donor tested. Non-parametric two-sided Kruskal-Wallis test corrected with Dunn's test for multiple comparisons was used to compare each group. **k** Immunoprecipitation (IP) of CTLA-4 in activated (+) or unstimulated (−) T cells from healthy donors. NBEAL2, LRBA, and CTLA-4 were immunoblotted on input fractions and on the IP fractions, as indicated (blots representative of $N = 3$ experiments). Source data are provided as a Source Data file.

GPS. These findings are suggestive of an overall immune dysregulation in NBEAL2 deficiency. Furthermore, the patients' neutrophils and eosinophils overexpressed activation markers (HLA-DR and FAS) and degranulation markers (CD66b) in the absence of stimulation and in the absence of documented infections. In particular, the overexpression of CD66b by neutrophils from patients with GPS confirms a previously report of abnormal activation and early degranulation[9]. These observations are consistent with the abnormally high plasma levels of neutrophil granule proteins measured in patients with GPS[8]. Our results corroborate these observations and extend them to eosinophils. The abnormally small size and low granularity of NBEAL2-deficient eosinophils[8] suggested that eosinophil functions might be perturbed in patients with GPS, albeit in the absence of overt clinical consequences. NK cells also had an activated profile, with the overexpression of CD11c, TIM3, and HLA-DR. Therefore, most granule-containing innate immune cells were perturbed in NBEAL2-deficient cells. However, the clinical consequences are generally moderate since recurrent infections (mostly of the upper respiratory tract) are only reported in a small proportion of patients with GPS[8]. It is noteworthy that the abnormal release of granule contents might contribute to the observed immune dysregulation in several ways[8,9,36]. Firstly, the excessive release of matrix metalloproteinases might contribute to the excessive cleavage of membrane-bound FasL and thus the elevated plasma concentration of soluble FasL[37]. However, given the large number of granule proteins with a broad spectrum of biological activity, the role of their excessive release fell outside the scope of the present work and would require dedicated investigation.

To better understand NBEAL2's role in T lymphocytes, we studied its interactome and identified its partners. A number of NBEAL2 partners have been previously described[10,24,25]. A mass spectrometry analysis was performed in a HEK cell line overexpressing only the pleckstrin homology, BEACH and WD40 repeat domains of NBEAL2[24]. The studies from Lo et al. revealed the interaction between NBEAL2 and SEC22B and P-selectin in megakaryocytes. In the present work, we identified NBEAL2's partners in activated primary T cells via pull-down of the endogenous protein. Furthermore, by comparing the interactomes in lymphocytes from healthy donors with lymphocytes from NBEAL2-deficient patients, we were able to assess the specificity of these interactions. Most of the NBEAL2-interacting proteins we identified by this strategy, are involved in cellular transport. NBEAL2 also interacts with proteins able to bind GTPases or cytoskeleton components. This is confirming the role of NBEAL2 as a scaffold protein in vesicular trafficking[13]. The expression of NBEAL2's partners in several immune cell subsets and in platelets emphasizes the broad range of the protein's potential functions. CCDC88B was one of the partner proteins with the highest intensity and control/patient ratio. Interestingly, murine Ccdc88b is known to bind to Dock8 and microtubules and (together with the dynactin/dynein complex) has a role in the transport of lytic granules along microtubules[28]. Consistently, NK cell degranulation is abnormal in *Nbeal2$^{-/-}$* mice[5]. Moreover, the presence of DYNC1H1 and GZMM (part of the dynein complex 1) in the interactome further suggests that NBEAL2 is a scaffold protein involved in lytic granule transport.

Lastly, we showed that NBEAL2 interacts with AP1G2 (the γ2 subunit of the adaptin protein complex AP-1) and LRBA[38]. Both LRBA and AP-1 are involved in CTLA-4 trafficking. Following endocytosis of CTLA-4, LRBA enables CTLA's recycling to the membrane[14], whereas the AP-1 complex mediates CTLA-4's transport to the lysosome for degradation[39]. Of note, CTLA-4 was not detected by mass spectrometry in NBEAL2-pull downs nor in proteome from activated T cells. The absence of CTLA-4 in the NBEAL2 interactome is likely due to technical limits. Indeed, another mass spectrometry study on sorted Treg and conventional T cells did not detect CTLA-4 in their dataset[40].

In view of NBEAL2's interaction with LRBA and AP1G2, we investigated the impact of NBEAL2 deficiency on CTLA-4 expression. After TCR activation, we observed a lower CTLA-4 expression, and thus less CTLA-4$^+$ conventional T cells, in GPS patients than in healthy controls. In contrast, NBEAL2 deficiency did not impair CTLA-4 expression on Tregs. This is consistent with the NBEAL2 expression profile at the protein level since we observed no, or very low, expression in Tregs from healthy donors. In contrast we found a strong NBEAL2 expression in activated T cells. These results therefore explain why the NBEAL2 deficiency lead to a defect in the expression of CTLA-4 on the conventional T cells but had no impact on the regulation of the expression of CTLA-4 in the Tregs. Our results indicate that despite normal CTLA-4 expression in Tregs, abnormally low CTLA-4 expression in a subset of conventional T cells can nevertheless be associated with the development of autoimmunity. Our observation of defective CTLA-4 expression in patients with GPS opens some new therapeutic options. Indeed, treatment with a CTLA-4-immunoglobulin (Ig) fusion protein showed good efficacy in patients with LRBA deficiencies and CTLA-4 haploinsufficiency[14,41]. Hence, the symptoms of GPS patients with autoimmune manifestations might also be alleviated by administration of CTLA-4-Ig to counteract the CTLA-4 defect observed in the effector T cells. Such a therapeutic approach could therefore be considered by clinicians taking care of GPS patients who present with autoimmune manifestations.

The single-cell RNA sequencing analysis showed that most pathways were similarly dysregulated in LRBA- and NBEAL2-deficient T cells, relative to healthy donor cells. However, the comparison between LRBA-deficient and NBEAL2-deficient cells revealed a specific upregulation of the STAT3/IL-6/JAK pathway in the memory T-cell subsets of patients with LRBA deficiency. Whether this signature reflects a lack of regulation of the memory T cells by Tregs as the consequence of LRBA deficiency remains to be confirmed in a larger group of patients. But it may constitute a new biomarker for Treg functional defects or to monitor the response to the CTLA-4-Ig treatment in LRBA patients.

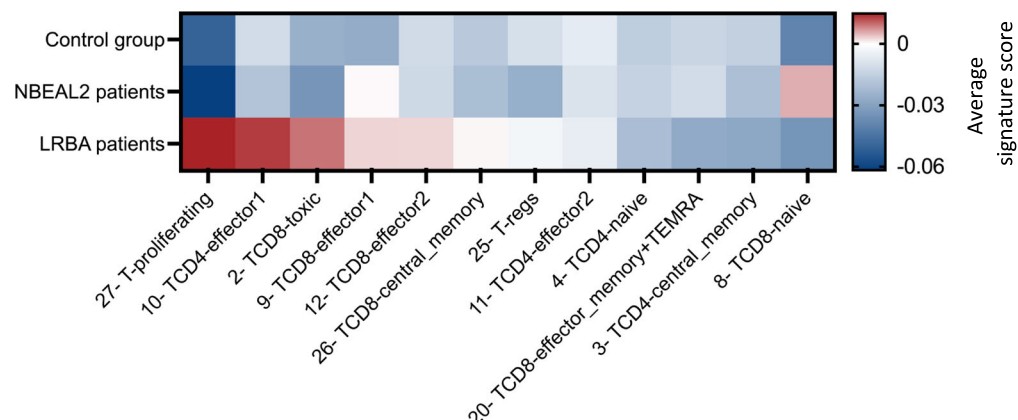

**a  T CD4+**

**b  T CD8+**

**c  STAT3/IL-6 signature**

Finally, the original finding that NBEAL2 is involved in CTLA-4 expression regulation in conventional T cells, but not in Tregs, suggest a distinct mechanism for the regulation of this immune checkpoint in these cell subsets, leading the way to more fundamental and therapeutics discoveries.

## Methods

This section briefly describes the methods. Further details are provided in the supplemental Methods and in supplementary tables 5–10. Informed consents and consents to publish the results have been obtained for all patients before the study. Before the

**Fig. 5 | Differential gene expression analyses: comparison of NBEAL2 and LRBA-deficient T cells. a** Differential Gene Expression (DEG) analyses between the NBEAL2-deficient cells ($N = 5$) or LRBA-deficient cells ($N = 5$) and the healthy controls group ($N = 7$). IPA Pathways analyses were performed on the T CD4+ lymphocytes subsets. In red the DEG correspond to the comparison between NBEAL2 patients and controls. In yellow, the DEG correspond to the comparison between LRBA patients and controls. **b** Differential Gene Expression (DEG) analyses between the NBEAL2 and LRBA patients' groups and the healthy controls group. IPA

Pathways analyses of the T CD8+ lymphocytes subsets. In red the DEG correspond to the comparison between NBEAL2 patients and controls. In yellow, the DEG correspond to the comparison between LRBA patients and controls. **c** Signature of the STAT3/IL-6 pathway in T cells subsets for the NBEAL2, LRBA and healthy controls groups. The heatmap shows the average signature score for the T cells subsets of each group. The list of genes from the signature Hallmark_IL6_JAK_-STAT3_signaling was used. Source data are provided as a Source Data file.

study, all patients signed informed consents approved by the CERAPH-Centre (IRB: #00011928). The biological samples are part of Inserm UMR1163/Imagine collection declared to the French Ministère de la recherche (CODECOH no. DC-2020-3994).

### CyTOF staining
Immune phenotyping on whole blood was carried out using the Maxpar Direct Immune Profiling kit (Fluidigm, Cat# 201325) with an antibody panel of 30 markers for CyTOF (Cytometry by Time Of Flight) analysis. To these 30 markers, 8 additional antibodies were added in order to detect FAS and certain immune checkpoints (TIM3, TIGIT, ICOS, GITR, PD-1, and CTLA-4). Samples were acquired on Helios mass cytometer.

### Immunoprecipitations
Anti-CTLA-4 (Abcam, cat# Ab251599) or anti-NBEAL2 (Abcam, Cat# ab250919) antibodies were coupled to agarose beads using the Thermo Pierce Direct IP kit (ThermoFisher, ref 26148) according to supplier's instructions. Prior to cells lysis with Octyl buffer, a reversible cross-linker was used to preserve protein-protein interactions: dithiobis[succinimidylproprionate] (DSP, Thermofisher, Cat# 22586). Lysates were incubated at 4 °C overnight with the beads coupled to the antibodies. After several washes of the beads on Pierce Spin Cups columns (Thermofisher, #69700), the immunoprecipitated fractions were recovered after denaturation with Blue loading buffer (CST, #56036S). Samples were reduced with β-mercaptoethanol (Biorad, ref #1610710) and analyzed either by immunoblotting or mass spectrometry.

### Mass spectrometry analysis
NBEAL2 immunoprecipitation fractions were separated by SDS-PAGE (4–20%) under reducing conditions. Gel bands were excised, reduced with DTT, alkylated with iodoacetamide and in-gel digested overnight with trypsin. Peptides were extracted with 50 mM ammonium bicarbonate and 50% acetonitrile in 0.2% formic acid, dried by evaporation in a speed-vac concentrator and resuspended in 0.2% formic acid. LC-MS/MS analyses were performed using a nano-ACQUITY Ultra-Performance LC system coupled to an Orbitrap Fusion Tribrid mass spectrometer.

### CRISPR knock-down
RNA guides to target NBEAL2 or LRBA were designed using the online tool CRISPOR. Primary activated T cells were electroporated with ribonucleoprotein complexes (Cas9 enzyme and the RNA guides) using the Amaxa™ 4D-Nucleofector nucleofaction system and the P2-EH100 program. After nucleofection, cells were cultured and restimulated with Dynabeads Human T-Activator CD3/CD28 at 1 beads/16 cells. Cells were stained to assess CTLA-4 expression by flow cytometry. Protein extractions were also performed.

### Single-cell RNA sequencing
Peripheral blood mononuclear cells (PBMC) from healthy donors, NBEAL2-deficient patients and LRBA-deficient patients were thawed. The scRNA-seq libraries were generated using Chromium Single Cell 3′ Library & Gel Bead Kit from 10x Genomics as previously described[42].

Data analyses were performed as described in the supplemental methods.

### Reporting summary
Further information on research design is available in the Nature Portfolio Reporting Summary linked to this article.

## Data availability
The new NBEAL2 variant sequence is deposited in National Center for Biotechnology Information. ClinVar; [VCV002443319.1], https://www.ncbi.nlm.nih.gov/clinvar/variation/VCV002443319.1. Single-cell RNA-sequencing data are available at the GEO accession number GSE196606. The mass spectrometry proteomics data have been deposited to the ProteomeXchange Consortium via the jPOST partner repository with the dataset identifier PXD042180 and JPST002157, which is publicly available. Interactome dataset visualized on STRING database is available here: https://version-11-5.string-db.org/cgi/network?networkId=bjY9nFy8le23. Source data are provided in this paper. Source data are provided with this paper.

## Code availability
As described in detail in the supplemental method section, CyTOF and single-cell RNA Seq data were analyzed using R software version 4.0.3 with the publicly available Catalyst and Seurat packages. Codes are available upon request to the corresponding author.

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

## Acknowledgements

We thank Jean-Claude Guillemot (Translational Sciences, Sanofi) for his help with mass spectrometry analyses; Sajedeh Sadat Mirshahvalad (INSERM UMR 1163, Institut *Imagine*) for the DNA purification prior to exome sequencing; Caroline Besnard (INSERM UMR 1163, Institut *Imagine*) for help with cytometry staining; and the staff from the Imagine genomic, bioinformatics, cytometry, and single-cell core facility and the Salpêtrière CyPS cytometry facility for advice and technical assistance. We sincerely thank the patients and their families for participating in the study. Lastly, we thank the attending physicians for their support and for coordinating the patients' clinical care and sample collection. This work was supported by several grants. LD and CdC received CIFRE PhD fellowships from Sanofi, France. LD also received a grant from the Imagine Thesis Award program. QR received an Institut Imagine MD-PhD fellowship (funded by the Fondation Bettencourt Schueller) and a Société Nationale Française de Médecine Interne (SNFMI) fellowship. SM received a grant from the INSERM and the Institut Imagine postdoctoral program (funded by the Fondation pour la Recherche Médicale; grant reference: FRM SPF20170938825). BPP received a postdoctoral fellowship from the ANRS (Agence nationale de recherches sur le sida). ASK was funded by an INTERREG V grant on autoimmune diseases and by Strasbourg University Medical Center. The LabTech Single-Cell@Imagine facility is funded by the Paris Region and the "Investissements d'avenir" program via the 2019 ATF – Sésame Filières PIA program (grant reference: 3877871). F.R.-L. lab was also funded by the Institut National de la Santé et de la Recherche Médicale (INSERM), the French government (managed by French National Research Agency (Agence Nationale de la Recherche) through the "Investissements d'avenir" program (Institut Hospitalo-Universitaire Imagine, grant reference: ANR-10-IAHU-01; Recherche Hospitalo-Universitaire, grant reference: ANR-18-RHUS-0010)), and other grants from the Agence National de la Recherche (ANR-18-CE17-0001 "Action" received by B.N.), the Fondation pour la Recherche Médicale (grant reference: Equipe FRM EQU202103012670), and the Centre de Référence Déficits Immunitaires Héréditaires (CEREDIH).

## Author contributions

Conceptualization by L.D., A.F., M.M., B.P. and F.R.-L.; Methodology by L.D., J-L.Z, F.C., Q.R., G.K., M.M., A.Magerus, B.P., and F.R.-L.; Investigation by L.D., J-L.Z., E.K., A.B., M-C.S, M.L, C.E., A. Michel. S.M., A.C., N.N., S.R., B.P.P., O.P., M.P., and A.Magerus; Data curation by L.D., F.C, Q.R., C.dC., A.B., M.B., A.Michel., and A.C.; Formal analysis by F.C., L.D., and

Q.R.; Writing – original draft by L.D.; Writing, review and editing by A.F., A-S.K., J-L.Z., B.P.P., C.dC., R.F., L.D., M.M., M-C.S., Q.R., B.P., and F.R-L.; Visualization by L.D., C.dC., F.C., Q.R., and J-L.Z.; Resources by B.N., A-S.K., R.F., B.C., G.K., S.H., G.L., S.L., M.F., C.L., M-C.S., and C.P.; Supervision by J-L.Z, B.P., and F.R.-L.; Funding acquisition by B.P. and F.R-L. All authors reviewed the manuscript.

## Competing interests

L.D., J-L.Z., E.K., A.B., G.K., C.E., S.R., and B.P. are employees of Sanofi, France. J-L.Z., E.K., A.B., G.K., C.dC., S.R., L.D., C.E., and B.P. may hold shares and/or stock options in the company. Other authors have nothing to disclose.

## Additional information

Laure Delage [1,2], Francesco Carbone [3,4,27], Quentin Riller [1,27], Jean-Luc Zachayus [5,27], Erwan Kerbellec [2], Armelle Buzy[6], Marie-Claude Stolzenberg[1], Marine Luka[3,4], Camille de Cevins[3,7], Georges Kalouche[8], Rémi Favier[9,10], Alizée Michel[1], Sonia Meynier[1], Aurélien Corneau [11], Caroline Evrard[5], Nathalie Neveux[12,13], Sébastien Roudières[6], Brieuc P. Pérot[3], Mathieu Fusaro[14], Christelle Lenoir[14], Olivier Pellé[1,15], Mélanie Parisot [16], Marc Bras[17], Sébastien Héritier[18], Guy Leverger[18], Anne-Sophie Korganow[19], Capucine Picard[20,21,22], Sylvain Latour[14], Bénédicte Collet[23], Alain Fischer [22,24,25], Bénédicte Neven[1,26], Aude Magérus[1], Mickaël Ménager [3,4], Benoit Pasquier[2,28] & Frédéric Rieux-Laucat [1,28] ✉

[1]Université Paris Cité, Institut Imagine, Laboratory of Immunogenetics of Pediatric Autoimmune Diseases, INSERM UMR 1163, F-75015 Paris, France. [2]Checkpoint Immunology, Immunology and Inflammation Therapeutic Area, Sanofi, F-94400 Vitry-sur-Seine, France. [3]Université Paris Cité, Institut Imagine, Laboratory of Inflammatory Responses and Transcriptomic Networks in Diseases, Atip-Avenir Team, INSERM UMR 1163, F-75015 Paris, France. [4]Labtech Single-Cell@Imagine, Imagine Institute, INSERM UMR 1163, F-75015 Paris, France. [5]Immunology and Inflammation Therapeutic Area, Sanofi, F-94400 Vitry-sur-Seine, France. [6]BioStructure and Biophysics, Integrated Drug Discovery, Sanofi, F- 94400 Vitry-sur-Seine, France. [7]Artificial Intelligence & Deep Analytics (AIDA) Group, Data & Data Science (DDS), Sanofi R&D, F- 91380 Chilly-Mazarin, France. [8]Cellomics, Translational Sciences, Sanofi, F- 91380 Chilly-Mazarin, France. [9]Assistance Publique-Hôpitaux de Paris, French national reference center for platelet disorders, Armand Trousseau Children Hospital, F-75012 Paris, France. [10]INSERM Unité Mixte de Recherche 1287, Gustave Roussy Cancer Campus, Paris-Saclay University, F-94805 Villejuif, France. [11]Sorbonne Université, UMS037, PASS, Plateforme de cytométrie de la Pitié-Salpêtrière CyPS, F-75013 Paris, France. [12]Laboratory of Biological Nutrition, EA 4466, Faculty of Pharmacy, Paris University, F-75014 Paris, France. [13]Clinical Chemistry Department, Hôpital Cochin, Assistance Publique - Hôpitaux de Paris (AP-HP), 4 Avenue de l'Observatoire, F-75014 Paris, France. [14]Université Paris Cité, Institut Imagine, Laboratory of Lymphocyte Activation and Susceptibility to EBV Infection, INSERM UMR 1163, F-75015 Paris, France. [15]Flow Cytometry Core Facility, Structure Fédérative de Recherche Necker, INSERM US24/CNRS UMS3633, F-75015 Paris, France. [16]Genomics Core Facility, Institut Imagine-Structure Fédérative de Recherche Necker, INSERM U1163 et INSERM US24/CNRS UAR3633, Université Paris Cité, F-75015 Paris, France. [17]Bioinformatics Platform, Structure Fédérative de Recherche Necker, INSERM UMR1163, Université Paris Cité, Imagine Institute, F-75015 Paris, France. [18]Sorbonne Université, INSERM UMRS_938, CRSA, AP-HP, Pediatric Oncology Hematology Unit, Hôpital Armand Trousseau, F-75012 Paris, France. [19]Department of Clinical Immunology and Internal Medicine, National Reference Center for Systemic Autoimmune Diseases (CNR RESO), Tertiary Center for Primary Immunodeficiency, Strasbourg University Hospital, F-67091 Strasbourg, France. [20]French National Reference Center for Primary Immune Deficiencies (CEREDIH), Necker-Enfants Malades University Hospital, AP-HP, F-75015 Paris, France. [21]Study Center for Primary Immunodeficiencies (CEDI), Necker-Enfants Malades University Hospital, AP-HP, F-75015 Paris, France. [22]Imagine Institute, INSERM UMR1163, Université Paris Cité, F-75015 Paris, France. [23]Pediatric Unit, Centre Hospitalier de Roubaix, F-59100 Roubaix, France. [24]Department of Paediatric Immuno-Haematology and Rheumatology, Reference Center for Rheumatic, AutoImmune and Systemic Diseases in Children (RAISE), Hôpital Necker-Enfants Malades, Assistance Publique – Hôpitaux de Paris (AP-HP), F-75015 Paris, France. [25]Collège de France, F-75231 Paris, France. [26]Pediatric Immunohematology and Rheumatology Department, Hôpital Necker-Enfants Malades, Assistance Publique-Hôpitaux de Paris (AP-HP), F-75015 Paris, France. [27]These authors contributed equally: Francesco Carbone, Quentin Riller, Jean-Luc Zachayus. [28]These authors jointly supervised this work: Benoit Pasquier, Frédéric Rieux-Laucat. ✉e-mail: frederic.rieux-laucat@inserm.fr

