## [Peer Review File · Nature Communications]

NBEAL2 deficiency in humans leads to low CTLA-4 expression in activated conventional T cellsREVIEWER COMMENTS

Reviewer #1 (Remarks to the Author):

In this manuscript the authors examine Grey Platelet Syndrome patients to learn how loss or dysfunction of NBEAL2 affects the immune responses. The specific focus of the study was the demonstration that loss of NBEAL2 function affected T-cells. Of specific note is that finding that CTLA-4 expression was affected. The manuscript reports a wide array of omics data. Of specific value is the interactome data. However, one is left with only a cloudy picture of just what NBEAL2 does in a T-cell (or any cell for that matter). Since NBEAL2 is thought to be an element of the cell's sorting machinery, an interesting analysis would be to look for discordance between expression and proteomics data to identify other proteins whose sorting/retention were disrupted without disruption of their transcription. Given the wealth of omics data reported, the authors could have easily done more to identify proteins that may need NBEAL2 for proper cellular localization and expression.

Additionally, such analysis might indicate if much of the protein over-expression as noted in their transcriptomics, was directly due to NBEAL2 dysfunction an indirect effect of something missorted. While the authors did address why GPS patients can have immunopathologies, it seems that an opportunity was missed to provide significant insights into the functions and mechanisms of this enigmatic protein.

Reviewer #2 (Remarks to the Author):

Comments for Authors:

Authors Delage L et al have submitted a manuscript for review titled: "NBEAL2 deficiency in humans leads to low CTLA-4 expression in conventional activated T cells". This work certainly expands the intellectual scope of its discovery beyond conventional medical subspecialties and underscores the importance of cross talk within fraternities of clinical hematology and basic immunology. Descriptive denomination of Gray Platelet Disease was coined by Giovanni Raccuglia in 1971 (G. Raccuglia Gray platelet syndrome: A variety of qualitative platelet disorder *Am J Med*, 51 (1971), pp. 818-828) based on platelet morphology as noted under the microscope in order to distinguish it from Glanzmann's Thrombasthenia. Gray Platelet Syndrome (GPS) is a consequence of loss of the α -granule storage pool of proteins in platelets. LRBA colocalized with CTLA4 in endosomal vesicles and LRBA deficiency or knockdown increased CTLA4 turnover, which resulted in reduced levels of CTLA4 protein in lymphocytes from patients with LRBA mutations (Lo et al, *Science* 2015). This manuscript connects the two clinical entities through work up the BEACH domain and extensive immunophenotyping of GPS patients and comparing their profiles with a cohort of LRBA patients.

• What are the noteworthy results?:

It has been known for some time (since 2011) that loss of function of a BEACH protein NBEAL2 leads to gray platelet syndrome (GPS) (per references 1 and 2) caused by Homozygosity/compound heterozygosity. Loss of function mutations in neurobeachin-like 2 (NBEAL2) is causative for Gray platelet syndrome (GPS; MIM #139090), characterized by thrombocytopenia and large platelets lacking α -granules and cargo. However, this work elegantly expands the understanding of its immunopathology and pathophysiology, and elegantly explains some intriguing aspects of the clinical presentations (autoimmune cytopenias, adenopathy, lymphopenia, thyroiditis, myelofibrosis and splenomegaly) by linking NBEAL2 function with CTLA4/LRBA trafficking. This work can also have therapeutic implications for treatment of patients with GPS.

• Will the work be of significance to the field and related fields? How does it compare to the established literature? If the work is not original, please provide relevant references.

Though some earlier work on similar lines has been cross referenced and discussed in this manuscript (references 7, 8, 9,10), one omission stands out: Fred G. Pluthero, Jorge DiPaola et al, *Platelets* 2018. In it Pluthero and colleagues posit: "NBEAL2 is a large gene with 54 exons, and several putative functional domains have been identified in NBEAL2, including PH (pleckstrin homology) and BEACH (beige and Chediak-Higashi) domains shared with other members of a protein family that includes LYST and LRBA, also expressed by hematopoietic cells". This article suggesting

link to other shared BEACH domains including LRBA in an earlier work by others should be mentioned and discussed.

• Does the work support the conclusions and claims, or is additional evidence needed?

Yes. Can authors kindly speculate/recommend beyond citing references 12-15, why some GPS patients with autoimmunity and autoinflammatory findings and end organ damage are candidates for therapy with CTLA4 immunoglobulin (Abatacept)?

• Are there any flaws in the data analysis, interpretation and conclusions? Do these prohibit publication or require revision? No, there are no flaws.

• Is the methodology sound? Does the work meet the expected standards in your field?

Yes. However, it is important to clarify some clinical details about the patients outlined in the Table in Figure1:

1. Only 1/13 patients seems to have platelet antibodies, did the authors measure the same for all patients ?

2. Similarly, splenomegaly due to extramedullary hematopoiesis is often a consequence of severe myelofibrosis. Hence both can coexist, it may be prudent to look at them separately and tabulate accordingly. Myelofibrosis has been recognized as a complication in some patients with GPS as early as 1978. (Caen et al. Bull Acad Natl Med 1991 Oct;175(7):1145-52; discussion 1152-3).

3. Any other sequelae of CTLA4/LRBA deficiency were notable: infiltrative lung, gut and brain lesions, endocrinopathies, increased PCR viral loads for Herpes family of viruses etc?

4. Why did one patient undergo alloBMT, did it help? We usually don't transplant GPS patients.

5. Is there any role for lysosomal toxins like Hydroxychloroquine in improving CTLA4 function in the GPS patients as a treatment modality ?

6. What are the biological implications if any of the STAT3-IL6 signature in GPS (Figure 5C)?

7. Same question about why and how to explain TREGs not being affected in GPS, compared to inherited CTLA4 haploinsufficiency patients.

8. Gamma Delta DNT cells were mentioned in Immunophenotype in Figure 2B. Can it be used as a good biomarker of immune dysregulation in GPS?

9. Any monocyte macrophage markers, RAS-ERK pathway markers were studied by CyTOF in GPS patients ?

• Is there enough detail provided in the methods for the work to be reproduced?

Yes. However, it will be critical for others in future to study by detailed immunophenotyping including activated lymphocyte CTLA4 expression in larger cohorts of patients with GPS. Hematologists that take care of GPS patients and clinical immunology laboratories need to collaborate for this as it has elegant therapeutic implications for these patients. Conversely qualitative platelet function including platelet morphology needs to be taken into account in patients with LRBA mutations, immune dysregulation and bleeding diathesis as some of those patients might benefit from simple interventions like DDAVP in an emergency. Overall, this manuscript challenges us to think critically beyond the tubular vision of our narrow clinical subspecialty silos.

V. Koneti Rao, MD, FRCPA.

Room 10/12C106

10, Center Drive

Bethesda, MD 20892-1899

<https://www.niaid.nih.gov/research/v-koneti-rao-md-frcpa>

Reviewer #3 (Remarks to the Author):

The paper by Delage, et.al., reports on the study of patients with grey platelet syndrome/ NBEAL2 deficiency. The manuscript reports on the extensive phenotyping and interactome analysis of NBEAL deficiency and conclude that this leads to impaired CTLA-4 expression in activated T cells and but not Treg.

Whilst the extended phenotyping and NBEAL interactome analysis is sensible enough and will appeal to some, it is just descriptive. The “key” finding and title of the paper relates to the potential control of CTLA-4 by NBEAL in activated T cells, but not Treg. This is a surprising and significant claim and therefore requires solid evidence which is rather lacking at present.

Major concerns:

1). The main figures (e.g. fig. 4) do not show any primary data supporting the low CTLA-4 expression in activated patient cells. The % CTLA-4 + is a strange measurement in this context and suggests that only ~10% of activated cells express CTLA-4. Since this is not normally the case one suspects either the activation or the staining is incorrect.

The authors argue it can't be activation, since CD25 expression is the equivalent. However they present no data on this, except figure S4D , which is presented as MFI. This is flawed as it is the % CD25 + cells that will help understand whether all cells are activated, showing the actual data is critical..

There is a continual problem of just showing % or MFI in bar graphs, where neither parameter captures the FACS data properly. The solution is for the authors to show actual CTLA-4 staining for all the patients (in supplementary) and put a representative example in the main figures. Preferably showing data for CTLA-4 Vs Foxp3, CTLA-4 vs CD25 etc in both resting and activated settings. As it stands there is very little actual data shown to support the title.

2) Whilst it is appreciated that patient samples may be limiting, the use of CRISPR to delete NBEAL2 provides an opportunity to study CTLA-4 expression in proper detail in edited healthy T cells. This should include staining at 37oC as a way of capturing CTLA-4 trafficking. The use of total CTLA-4 staining in fixed cells as used presently is limiting, since it seems more likely that NBEAL should affect trafficking, given its association with LRBA.

3). It seems strange that an NBEAL defect should affect the % CTLA-4 + cells after activation rather than level of CTLA-4 expression itself (which according to S4C is the case). Again more carefully controlled experiments, presented with appropriate gating strategies to show any impact on naïve vs memory fractions, activated vs not activated, FoxP3+ vs non Foxp3 etc. should all be possible to make the authors case much stronger. As it stands the simplest explanation is that the CTLA-4 staining and activation conditions may be suboptimal and Treg CTLA-4 staining is resistant to these issues due to their normally higher expression of CTLA-4. The fact that S4B suggest that Treg express lower levels than conventional T cells is a concern.

Reviewer #4 (Remarks to the Author):

As a special reviewer, I'd only provide comments on the session of scRNAseq study of this manuscript as follows:

1. scRNAseq quality data of all patient samples are needed to verify if the results were unbiased or driven by data quality. These include the number of cells before and after filtering, percentages of reads in cells, cell viabilities after PBMC were thaw, labeling patient IDs in UMAPs, plotting gene coverages/total UMI counts per cell/percentages of mitochondria genes in UMAPs.
2. The authors identified and annotated 35 clusters after data integration, which was a concern. The data integration step commonly causes over-correction of data that force different cell populations into same clusters. I'd suggest identifying cell sub-type patient by patient, followed by data integration. The prior-defined cell types should be used to guide data integration, i.e., same cell types should be clustered together after data integration by tuning the integration methods/parameters.
3. Please provide evidence of the annotation of the 35 clusters, i.e., expression values of known markers of each cell type.
4. Ambient RNAs are commonly seen in scRNAseq of frozen samples. Please try to remove ambient RNAs using tools such as SoupX and compare results before and after removal of ambient RNAs.
5. DEG analysis used log₂FC cutoff of 0, which corresponds to fold change of 1. I'd suggest using log₂FC cutoff of 1 (equivalent fold change of 2) or 0.58 (equivalent fold change of 1.5).

REVIEWER COMMENTS

Reviewer #1 (Remarks to the Author):

In this manuscript the authors examine Grey Platelet Syndrome patients to learn how loss or dysfunction of NBEAL2 affects the immune responses. The specific focus of the study was the demonstration that loss of NBEAL2 function affected T-cells. Of specific note is that finding that CLTA-4 expression was affected. The manuscript reports a wide array of omics data. Of specific value is the interactome data. However, one is left with only a cloudy picture of just what NBEAL2 does in a T-cell (or any cell for that matter). Since NBEAL2 is thought to be an element of the cell's sorting machinery, an interesting analysis would be to look for discordance between expression and proteomics data to identify other proteins whose sorting/retention were disrupted without disruption of their transcription. Given the wealth of omics data reported, the authors could have easily done more to identify proteins that may need NBEAL2 for proper cellular localization and expression. Additionally, such analysis might indicate if much of the protein over-expression as noted in their transcriptomics, was directly due to NBEAL2 dysfunction an indirect effect of something missorted. While the authors did address why GPS patients can have immunopathologies, it seems that an opportunity was missed to provide significant insights into the functions and mechanisms of this enigmatic protein.

We thank the reviewer for the enthusiastic comments about the wealth and value of our interactome.

In addition to the interactome, we analyzed the whole proteome of two NBEAL2 patients versus two healthy donors (HDs). This analysis was performed on the input fractions (before immunoprecipitation), to confirm that the specific NBEAL2 partners identified were equivalently expressed in patients and controls.

As suggested by reviewer #1, we came back to our proteome analyses. After stringent selection, we identified 21 proteins expressed in HDs but not in NBEAL2-tested patients, 23 proteins expressed in patients but not in HDs and 4 proteins with at least 2-fold intensities differences (48 proteins in total). The table has been added in the supplemental material (Table S4). Overexpressed proteins in tested GPS patients are mainly involved in the regulation of TP53 activity and transcriptional activity. Under-expressed proteins in the tested GPS patients are mainly involved in metabolic pathways.

To be able to compare our proteomic data with transcriptomic data, we performed bulk RNA sequencing on activated T cells. Indeed, in our first version of the article, we provided single-cell RNA sequencing on PBMC and T cells without in vitro activation. After DEseq2 analysis of the bulk RNA sequencing, we found very few differentially expressed genes when we compared NBEAL2 patients' T cells to healthy donors (HD). None of them were the transcripts of the proteins previously identified in the proteome. The normalized counts are given in the second part of the Table S4.

Using only proteomics/transcriptomics data on activated T cells, it is uneasy to conclude if the expression of the dysregulated proteins is related to the missorting of the proteins or another mechanism. Moreover, the correlation between transcriptomic and protein expression is not always linear and strongly depends on the RNA quantification methods. It was estimated in cell lines that there is a good correlation for only a third of the genes (Gry et al BMC Genomics

2009, 10:365 doi:10.1186/1471-2164-10-365). It could therefore be speculative to draw a conclusion on the discrepancy between the abundance of transcripts and that of the corresponding proteins.

In this study, we focused on the proteins interacting with NBEAL2. In this regard, our interactome provides an overview of the proteins that interact specifically with NBEAL2 in activated T cells. Since we identified LRBA as a specific partner and given its role in the intracellular traffic of CTLA-4, we thus focused our study on the CTLA-4 expression in T cells. Indeed, a reduced CTLA-4 expression in human T cells is strongly associated with the onset of autoimmune manifestations, a known but so far unexplained complication in GPS patients. Our goal was not to decipher the fundamental biology of NBEAL2 in T cells. This would require extensive additional experiments and would constitute a project in itself. We are also convinced there is more to discover about NBEAL2 functions and mechanisms. Hence, sharing our omics data with the scientific community will enrich the field and probably lead to exciting discoveries in the future.

Reviewer #2 (Remarks to the Author):

Comments for Authors:

Authors Delage L et al have submitted a manuscript for review titled: "NBEAL2 deficiency in humans leads to low CTLA-4 expression in conventional activated T cells". This work certainly expands the intellectual scope of its discovery beyond conventional medical subspecialties and underscores the importance of cross talk within fraternities of clinical hematology and basic immunology. Descriptive denomination of Gray Platelet Disease was coined by Giovanni Raccuglia in 1971 (G. Raccuglia Gray platelet syndrome: A variety of qualitative platelet disorder *Am J Med*, 51 (1971), pp. 818-828) based on platelet morphology as noted under the microscope in order to distinguish it from Glanzmann's Thrombasthenia. Gray Platelet Syndrome (GPS) is a consequence of loss of the α -granule storage pool of proteins in platelets. LRBA colocalized with CTLA4 in endosomal vesicles and LRBA deficiency or knockdown increased CTLA4 turnover, which resulted in reduced levels of CTLA4 protein in lymphocytes from patients with LRBA mutations (Lo et al, *Science* 2015). This manuscript connects the two clinical entities through work up the BEACH domain and extensive immunophenotyping of GPS patients and comparing their profiles with a cohort of LRBA patients.

• What are the noteworthy results?:

It has been known for some time (since 2011) that loss of function of a BEACH protein NBEAL2 leads to gray platelet syndrome (GPS) (per references 1 and 2) caused by Homozygosity/compound heterozygosity. Loss of function mutations in neurobeachin-like 2 (NBEAL2) is causative for Gray platelet syndrome (GPS; MIM #139090), characterized by thrombocytopenia and large platelets lacking α -granules and cargo. However, this work elegantly expands the understanding of its immunopathology and pathophysiology, and elegantly explains some intriguing aspects of the clinical presentations (autoimmune cytopenias, adenopathy, lymphopenia, thyroiditis, myelofibrosis and splenomegaly) by linking NBEAL2 function with CTLA4/LRBA trafficking. This work can also have therapeutic implications for treatment of patients with GPS.

We thank Pr Rao for his enthusiastic comments and the transparency of his review.

• Will the work be of significance to the field and related fields? How does it compare to the established literature? If the work is not original, please provide relevant references.

Though some earlier work on similar lines has been cross referenced and discussed in this manuscript (references 7, 8, 9,10), one omission stands out: Fred G. Pluthero, Jorge DiPaola et al, Platelets 2018. In it Pluthero and colleagues posit: “NBEAL2 is a large gene with 54 exons, and several putative functional domains have been identified in NBEAL2, including PH (pleckstrin homology) and BEACH (beige and Chediak-Higashi) domains shared with other members of a protein family that includes LYST and LRBA, also expressed by hematopoietic cells”. This article suggesting link to other shared BEACH domains including LRBA in an earlier work by others should be mentioned and discussed.

We acknowledge that this reference was missing. We now added this reference to our manuscript (line 111). Nevertheless, we would like to mention that this article by Pluthero and colleagues mentioned that NBEAL2 is a member of the BEACH family proteins, like LRBA and LYST. However, the authors did not develop further on the similarities between these proteins and, more importantly, did not speculate on a possible link between NBEAL2 and CTLA-4. However, as above-mentioned, we agree that it is fair to reference this article.

• Does the work support the conclusions and claims, or is additional evidence needed?

Yes. Can authors kindly speculate/recommend beyond citing references 12-15, why some GPS patients with autoimmunity and autoinflammatory findings and end organ damage are candidates for therapy with CTLA4 immunoglobulin (Abatacept)?

As proposed by Pr Rao, we have added a sentence to encourage clinicians taking care of GPS patients with autoimmune manifestations to consider using abatacept as a possible treatment in these patients (lines 378 -381).

• Are there any flaws in the data analysis, interpretation and conclusions? Do these prohibit publication or require revision? No, there are no flaws.

• Is the methodology sound? Does the work meet the expected standards in your field?

Yes. However, it is important to clarify some clinical details about the patients outlined in the Table in Figure1:

1. Only 1/13 patients seems to have platelet antibodies, did the authors measure the same for all patients ?

Monoclonal antibody-specific immobilization of platelet antigens (MAIPA) assays were performed for patients P2, families 4 and 5, P6, P7 and P8. Unfortunately, patients P1, P3, and family 9 have not been tested.

We sincerely hope that our work, as well as other studies like the international cohort (Sims et al., Blood, 2021), will increase the use of standardized tests to search for autoantibodies in patients suffering from GPS.

2. Similarly, splenomegaly due to extramedullary hematopoiesis is often a consequence of severe myelofibrosis. Hence both can coexist, it may be prudent to look at them separately and tabulate accordingly. Myelofibrosis has been recognized as a complication in some patients with GPS as early as 1978. (Caen et al. Bull Acad Natl Med 1991 Oct;175(7):1145-52; discussion 1152-3).

Thanks to Pr Rao’s (reviewer #2) comment, we have now tabulated separately “myelofibrosis” and “splenomegaly” in Figure 1.A.

3. Any other sequelae of CTLA4/LRBA deficiency were notable: infiltrative lung, gut and brain lesions, endocrinopathies, increased PCR viral loads for Herpes family of viruses etc?

If we consider the patients described in the present study, and those described by Sims et al. in Blood, no digestive or cerebral manifestations have been reported so far. In contrast, autoimmune thyroiditis has been reported (patient P5.II-1 in our study and P22.1 in Sims et al., Blood, 2021).

To our knowledge, only one pulmonary fibrosis was once described in a GPS patient (Facon, Thierry, et al. "Simultaneous occurrence of grey platelet syndrome and idiopathic pulmonary fibrosis: a role for abnormal megakaryocytes in the pathogenesis of pulmonary fibrosis?." British journal of haematology 74.4 (1990): 542-543.. Some GPS patients have been reported with recurrent infections (Chedani H et al., Platelets. 2006;17(1):14-19 ; Drouin A et al., Blood. 2001;98(5): 1382-1391 ; Kahr WH et al., Blood. 2012;120(13):2543.). Regarding viral loads for herpes family viruses, one patient out of 13 (P3) developed several episodes of EBV reactivation as indicated in table 1. Remission was obtained after anti-CD20 therapy. After B cells recovery, a mild hypo Ig (6mg/dl) persisted without related infection. This information was added in the main text (lines 151 to 154).

4. Why did one patient undergo alloBMT, did it help? We usually don't transplant GPS patients.

Indeed, transplantation is not a standard of care for GPS patients. Nevertheless, this patient presented with severe pancytopenia, so bone marrow transplantation (BMT) was performed in emergency. This specific clinical case is fully described in the paper (Favier, Rémi, et al. "Correction of severe myelofibrosis, impaired platelet functions and abnormalities in a patient with gray platelet syndrome successfully treated by stem cell transplantation." Platelets 31.4 (2020): 536-540.), as cited in our manuscript. After BMT, the patient improved and is now in complete remission. Of note, as explained in the text, this patient has been excluded from our cohort.

5. Is there any role for lysosomal toxins like Hydroxychloroquine in improving CTLA4 function in the GPS patients as a treatment modality ?

This is a very interesting question, as hydroxychloroquine (HCQ) is known to impact lysosomal acidification, thereby reducing lysosomal CTLA-4 destruction. Consequently, an HCQ treatment could possibly restore some CTLA-4 expression at the cell surface as previously described (Lo et al. science 2015, PMID26206937). However, this impact remains modest, and finally, HCQ therapy is not a standard of care in LRBA patients, nor in GPS patients so far. To our knowledge, there is no clinical trial published that properly assessed the HCQ efficacy in LRBA or CTLA-4 patients. Therefore, the abatacept therapy remains the standard of care in those patients. Considering that the reduced CTLA-4 expression in GPS patients is restricted to conventional T cells, one could speculate that HCQ might have a sufficient impact. Nevertheless, it could be challenging to assess the HCQ therapy in children given that the abatacept is a well-proven efficient therapy. Eventually, this could be an option in some adult LRBA or CTLA-4 patients with autoimmune enteropathy refractory to the abatacept therapy. This could be considered on a case-by-case basis, and therefore it did not seem justified to consider it here.

6. What are the biological implications if any of the STAT3-IL6 signature in GPS (Figure 5C)?

We are particularly grateful to Pr Rao for this question. Indeed, this first allows us to emphasize that we observed very little transcriptional difference between the cells of NBEAL2 and LRBA patients, thus suggesting that the deregulations are very similar. This is why we wanted to share

the identification of this IL6-STAT3 signature which seems quite unique. It is not easy to draw a definitive conclusion. However, we can speculate on the significance of this signature which could be associated with a defect in the regulation of conventional LTs by Tregs, as it was observed in LRBA patients. If confirmed in additional LRBA or CTLA-4 patients, it could be potentially used as a biomarker to monitor the abatacept therapy in those patients as discussed (Lines 388-390).

7. Same question about why and how to explain TREGs not being affected in GPS, compared to inherited CTLA4 haploinsufficiency patients.

This is indeed a key question in this study. Our transcriptional analyses (Fig. S7B) and transcriptomic profiles from public databases suggested that NBEAL2 is poorly expressed in Tregs at the RNA level. However, there might be non-linear relation between the RNA and protein expression. To bring a molecular rationale to our results, we measured, at the protein level, the expression of NBEAL2 and LRBA in Tregs and conventional CD4⁺CD25⁻ T cells (Tconv) from 3 healthy donors (Fig S6). Total proteins were extracted from sorted Tregs and Tconv as well as from *in vitro* activated Tconv. Very interestingly, NBEAL2 is not, or faintly, detected in Tregs, but it is well expressed in Tconv (Fig S6d and e). Importantly, we observed an increased NBEAL2 expression in activated Tconv. For LRBA, we observed a mirror expression profile compared to NBEAL2. LRBA expression is high in Tregs and T conv, but its expression is weaker in *in vitro* activated Tconv.

Overall, these new results show that NBEAL2 is not, or very weakly, expressed in Tregs. This observation is, therefore, consistent with the normal expression of CTLA-4 in the Tregs of NBEAL2-deficient GPS patients, and suggests that in the Tregs, LRBA is the main regulator of CTLA-4 trafficking. In Tconv the picture looks more complex. Indeed, if we observed a decreased LRBA expression in the activated compared to the non-activated Tconvs, there remains a notable expression of LRBA. This indicates that the co-expression of NBEAL2 and LRBA might be necessary in Tconvs to ensure the recycling of CTLA-4 to a normal level. This is also consistent with our results obtained in NBEAL2 or LRBA deletion experiments using the CRISPR/Cas9 approach in primary activated T cells from healthy donors. A detailed biochemical study would be necessary to understand better the interactions and the precise mechanisms by which NBEAL2 and LRBA coregulate the intracellular trafficking of CTLA-4 in T conv.

8. Gamma Delta DNT cells were mentioned in Immunophenotype in Figure 2B. Can it be used as a good biomarker of immune dysregulation in GPS?

To generalize our finding, an assessment on a larger cohort of patients is warranted. Moreover, the proportion of this minor T cell subset is fluctuating upon infections. Therefore, we do not think that gamma delta DNT cells can be used as a robust biomarker of immune dysregulation in GPS since.

9. Any monocyte macrophage markers, RAS-ERK pathway markers were studied by CyTOF in GPS patients ?

Commonly used monocyte or macrophage markers were studied by CyTOF, including CD14 CD16, and HLA-DR. Unfortunately, RAS-ERK pathway markers were not included in our panel.

- Is there enough detail provided in the methods for the work to be reproduced?

Yes. However, it will be critical for others in future to study by detailed immunophenotyping including activated lymphocyte CTLA4 expression in larger cohorts of patients with GPS. Hematologists that take care of GPS patients and clinical immunology laboratories need to collaborate for this as it has elegant therapeutic implications for these patients. Conversely qualitative platelet function including platelet morphology needs to be taken into account in patients with LRBA mutations, immune dysregulation and bleeding diathesis as some of those patients might benefit from simple interventions like DDAVP in an emergency. Overall, this manuscript challenges us to think critically beyond the tubular vision of our narrow clinical subspecialty silos.

We would like to particularly thank Pr Rao for his constructive remarks, which improved our manuscript. Moreover, we share with him the notion that we must try as much as possible to open our thoughts to the clinical manifestations that are not always in the foreground in the context of a given genetic deficit, to bring to these patients the most specific and effective treatments.

V. Koneti Rao, MD, FRCPA.
Room 10/12C106
10, Center Drive
Bethesda, MD 20892-1899
<https://www.niaid.nih.gov/research/v-koneti-rao-md-frcpa>

Reviewer #3 (Remarks to the Author):

The paper by Delage, et.al., reports on the study of patients with grey platelet syndrome/NBEAL2 deficiency. The manuscript reports on the extensive phenotyping and interactome analysis of NBEAL deficiency and conclude that this leads to impaired CTLA-4 expression in activated T cells and but not Treg.

Whilst the extended phenotyping and NBEAL interactome analysis is sensible enough and will appeal to some, it is just descriptive. The “key” finding and title of the paper relates to the potential control of CTLA-4 by NBEAL in activated T cells, but not Treg. This is a surprising and significant claim and therefore requires solid evidence which is rather lacking at present.

We thank reviewer #3 for his comments and recognition of the interest in this study which brings new knowledge to the function of NBEAL2 in T cells. We agree that the control of CTLA-4 by NBEAL2 in conventional T cells but not in Tregs is a significant and new finding. We performed new experiments, which are now presented in our manuscript’s revised version, and we believe these results confirm and consolidate our initial results. We measured, at the protein level, the expression of NBEAL2 and LRBA in Tregs and conventional CD4+CD25- T cells (Tconv) from 3 healthy donors. Total proteins were extracted from sorted Tregs and Tconv as well as from *in vitro* activated Tconv. Very interestingly, NBEAL2 is not, or faintly, detected in Tregs, but it is well expressed in Tconv (Fig S6 d and e). Importantly, we observed an increased NBEAL2 expression in activated Tconv. For LRBA, we observed a mirror expression profile compared to NBEAL2. LRBA expression is high in Tregs and T conv, but its expression is weaker in *in vitro* activated Tconv.

Overall, these new results show that NBEAL2 is not, or very weakly, expressed in Tregs. This observation is, therefore, consistent with the normal expression of CTLA-4 in the Tregs of

NBEAL2-deficient GPS patients. It also suggests that in Tregs, LRBA might be the main regulator of CTLA-4 trafficking, whereas NBEAL2 is dispensable.

Major concerns:

1). The main figures (e.g. fig. 4) do not show any primary data supporting the low CTLA-4 expression in activated patient cells. The % CTLA-4 + is a strange measurement in this context and suggests that only ~10% of activated cells express CTLA-4. Since this is not normally the case one suspects either the activation or the staining is incorrect.

The authors argue it can't be activation, since CD25 expression is the equivalent. However they present no data on this, except figure S4D, which is presented as MFI. This is flawed as it is the % CD25 + cells that will help understand whether all cells are activated, showing the actual data is critical. There is a continual problem of just showing % or MFI in bar graphs, where neither parameter captures the FACS data properly. The solution is for the authors to show actual CTLA-4 staining for all the patients (in supplementary) and put a representative example in the main figures. Preferably showing data for CTLA-4 Vs Foxp3, CTLA-4 vs CD25 etc in both resting and activated settings. As it stands there is very little actual data shown to support the title.

We respectfully disagree with the reviewer's conclusion regarding T cell activation. As requested, we added the percentage of CD25⁺ cells, and showed the histogram profiles of this staining in supplemental figure S5. The vast majority of T cells are CD25⁺ upon in vitro activation. These data are thus demonstrating that the in vitro activation was similarly efficient in T cells from patients and healthy donors.

We decided to show a percentage of CTLA-4 positive cells rather than the MFI of CTLA-4 staining because the antibody used as isotype control had elevated MFI as well. However, if we focus on CTLA-4 MFI of the entire activated T cell subset, the MFI between patients and controls can be compared. We have normalized the MFI of each sample by the mean MFI of the internal control (healthy donors of the day), therefore performing the following calculation: $MFI_{\text{sample}} / MFI_{\text{mean_controls}}$. We now show the normalized CTLA-4 MFI values on Tregs and activated T cells from patients and healthy donors (Fig. 4c and e).

The dot plots were already present in the previous figure S4E and were now moved to figure 4d for some patients to give an example, as requested by reviewer#3. Lastly, we added the dot plots for all patients in supplemental figure S4 (for the Tregs) and S5 (for the activated T cells).

2) Whilst it is appreciated that patient samples may be limiting, the use of CRISPR to delete NBEAL2 provides an opportunity to study CTLA-4 expression in proper detail in edited healthy T cells. This should include staining at 37°C as a way of capturing CTLA-4 trafficking. The use of total CTLA-4 staining in fixed cells as used presently is limiting, since it seems more likely that NBEAL should affect trafficking, given its association with LRBA.

The availability of patient cells is indeed limited, but the generation of donor lymphocytes by CRISPR editing is also limited. We could generate enough cells to carry out a WB and phenotyping. However, it would be necessary to be able to obtain many more cells in order to carry out a study of the intracellular traffic of CTLA-4. Our study aimed to understand the possible mechanisms that lead to autoimmunity in GPS patients. Highlighting the interaction between NBEAL2 and LRBA led us to focus our study on the expression of CTLA-4, given the known role of LRBA in the regulation of CTLA-4. Our results indicate that NBEAL2 deficiency has a similar impact to LRBA but only in conventional T cells. This editing approach aimed to confirm this observation in healthy donor cells to ensure that this phenomenon is the consequence of the NBEAL2 defect and not a phenomenon only observed in patient cells. This

discovery will pave the way for projects aimed at understanding the precise mechanisms, but we respectfully believe this goes beyond our study.

3). It seems strange that an NBEAL defect should affect the % CTLA-4 + cells after activation rather than level of CTLA-4 expression itself (which according to S4C is the case). Again more carefully controlled experiments, presented with appropriate gating strategies to show any impact on naïve vs memory fractions, activated vs not activated, FoxP3+ vs non Foxp3 etc. should all be possible to make the authors case much stronger. As it stands the simplest explanation is that the CTLA-4 staining and activation conditions may be suboptimal and Treg CTLA-4 staining is resistant to these issues due to their normally higher expression of CTLA-4. The fact that S4B suggest that Treg express lower levels than conventional T cells is a concern.

We agree with the reviewer that it could be a concern if Tregs would express less CTLA-4 than Tconv. It is not possible to conclude from the MFI values of the previous Figure S4B that Tregs express lower levels of CTLA-4 than conventional T cells. There are several reasons for it, as it was detailed in the methods. Firstly, we used different antibody panels for the staining of Tregs and Tconvs, and in particular, the anti-CTLA-4 antibody was coupled to different fluorochromes in the Tregs and the Tconvs panels. Secondly, those staining experiments were acquired on different cytometers. Therefore, it is not accurate to compare the MFIs in the different experiments to infer that there is less CTLA-4 in Tregs than in Tconv. With revisions and the normalized MFI values these figures, possibly misleading, have been removed.

As pointed out by reviewer#3, our results indicated that there are fewer CTLA-4+ conventional T cells in the patients than in healthy controls. We confirmed this observation in NBEAL2-edited activated T cells from healthy donors. We have revised our analyses accordingly, and we are now showing a reduced CTLA-4 expression in activated Tconvs from the patients but not in their Tregs. In addition, we confirmed this observation in NBEAL2 KD-activated T cells from healthy donors.

As requested by reviewer #3, to further confirm that the NBEAL2 deficiency has no impact in Tregs, we are now showing the CTLA-4 expression in Tregs from patients and healthy donors, using either a CD4⁺ CD127_{low} CD25⁺ or a CD4⁺FOXP3⁺HELIOS⁺ gating (Fig S4). As shown in this figure, we observed some inter-individual variations with regard to the CTLA-4 MFI, even in healthy donors. But there is no significant difference between patients and controls, either for the proportion of Tregs, the level of CTLA-4 expression, or the number of CTLA-4 positive Tregs. These results are consistent with the WB analysis showing barely detectable NBEAL2 expression in Tregs (Fig. S6). Therefore, the NBEAL2 deficiency has no impact on CTLA-4 expression in Tregs.

Reviewer #4 (Remarks to the Author):

As a special reviewer, I'd only provide comments on the session of scRNAseq study of this manuscript as follows:

1. scRNAseq quality data of all patient samples are needed to verify if the results were unbiased or driven by data quality. These include the number of cells before and after filtering, percentages of reads in cells, cell viabilities after PBMC were thaw, labeling patient IDs in

UMAPs, plotting gene coverages/total UMI counts per cell/percentages of mitochondria genes in UMAPs.

We performed quality controls in all samples before integration. We compiled all the data requested in a new supplemental table 3.

We also provide UMAPs for each patient in supplemental figures 8 and 10.

2. The authors identified and annotated 35 clusters after data integration, which was a concern. The data integration step commonly causes over-correction of data that force different cell populations into same clusters. I'd suggest identifying cell sub-type patient by patient, followed by data integration. The prior-defined cell types should be used to guide data integration, i.e., same cell types should be clustered together after data integration by tuning the integration methods/parameters.

We thank the reviewer for his guidance. The method s/he is proposing is not feasible owing to the number of samples analyzed.

The over-correction of the data is not of concern because the anchor between the samples was found using reciprocal PCA (rPCA). In rPCA, each sample is projected into the other PCA spaces, and it constrains the anchors by the same mutual neighborhood requirement. In addition to being faster than other methods (e.g., cca), it is also a conservative approach in which the biological states are less likely to 'align' after integration.

(https://satijalab.org/seurat/articles/integration_rpca.html)

Once the samples are integrated, a two-step process is used to annotate the dataset.

Initially, a broad annotation is done by transferring the labels from an already annotated high-quality reference dataset (data not shown) using the MapQuery function in Seurat. This initial annotation is then manually refined using a set of markers for every cell type available in peripheral mononuclear blood cells (PBMC).

The level and the detail of the resulting annotation are usually project-based. In this case, being the T cells the focus of the analysis, it was decided to keep all the clusters and avoid merging them (e.g., 9-TCD8-effector1 and 12-TCD8-effector2).

We are confident that the proposed method provides an accurate and detailed description of the cell profile of the dataset. This is confirmed by the good correlation between this analysis and the percentage of cells obtained from mass cytometry.

As a quality control and to make sure there is no overcorrection of the data, we are also providing the UMAP per sample/patient (see supplemental material, Fig S8).

3. Please provide evidence of the annotation of the 35 clusters, i.e., expression values of known markers of each cell type.

In supplemental figure 9, we added a heatmap of selected markers used to identify each cell cluster.

4. Ambient RNAs are commonly seen in scRNAseq of frozen samples. Please try to remove ambient RNAs using tools such as SoupX and compare results before and after removal of ambient RNAs.

We run SoupX on our samples as asked by the reviewer.

The level of ambient mRNA present in the samples was calculated using soupX with default settings. The raw and filtered matrices from Cellranger were loaded for each sample using the load10X function. The contamination fraction was calculated by combining the results of the

automated method (autoEstCont function) with the manual one, in which immunoglobulin genes and other genes linked to contamination (PPBP, HBB, and HBA1) were passed to estimateNonExpressingCells to estimate the contamination values. All the samples had contamination levels lower than the expected contamination range for frozen samples (10%-40%, as reported by SoupX's authors). No correction was applied. The results are in Supplemental table 3.

5. DEG analysis used log₂FC cutoff of 0, which corresponds to fold change of 1. I'd suggest using log₂FC cutoff of 1 (equivalent fold change of 2) or 0.58 (equivalent fold change of 1.5).

We thank the reviewer for this comment. Indeed, we initially generated the DEG analysis using a log₂FC cut-off of 0. However, when we used Ingenuity Pathway Analysis (IPA), we set the cut-off at a log₂FC of 0.25 (1.2-fold change). This cut-off was previously used in other published works (de Cevins, Camille, et al. *Med* 2.9 (2021): 1072-1092 and Riller, Quentin, et al. *Journal of Allergy and Clinical Immunology* (2023)). We corrected and specified this cut-off in the supplemental data material and methods section (lines 316-317).

REVIEWERS' COMMENTS

Reviewer #1 (Remarks to the Author):

The authors have addressed my concerns and improved the manuscript and its value to the field. I think this paper will have an impact in its revised form.

Reviewer #2 (Remarks to the Author):

No I have no further comments or concerns for this manuscript. Authors have adequately answered my concerns.

Reviewer #3 (Remarks to the Author):

The authors have not provided some of the raw data underpinning their claims. From this it emerges that CTLA-4 expression was measured using different fluorochromes on different panels for Treg and non Treg. This is a concern to me as there is no direct way to make a comparison between these subsets by flow. Fundamentally my view is that their CTLA-4 staining is poor since all activated T cells will express this protein and yet the authors only observe a small %. The normalising of values to control samples is problematic since there is significant variation depending on the control used. Therefore my feeling is their main conclusion remains somewhat questionable.

Nonetheless there is still interesting information in the paper. the fact that NBEAL2 appears to be absent from Treg is a surprise and may explain some aspects of the data, but equally it makes it hard to understand the relationship between LRBA and NBEAL2 in terms of redundancy / essentiality for CTLA-4 control.

Reviewer #4 (Remarks to the Author):

All my comments have been addressed.

REVIEWERS' COMMENTS

Reviewer #1 (Remarks to the Author):

The authors have addressed my concerns and improved the manuscript and its value to the field. I think this paper will have an impact in its revised form.

Reviewer #2 (Remarks to the Author):

No I have no further comments or concerns for this manuscript. Authors have adequately answered my concerns.

Reviewer #3 (Remarks to the Author):

The authors have now provided some of the raw data underpinning their claims. From this it emerges that CTLA-4 expression was measured using different fluorochromes on different panels for Treg and non Treg. This is a concern to me as there is no direct way to make a comparison between these subsets by flow. Fundamentally my view is that their CTLA-4 staining is poor since all activated T cells will express this protein and yet the authors only observe a small %. The normalizing of values to control samples is problematic since there is significant variation depending on the control used. Therefore my feeling is their main conclusion remains somewhat questionable.

Nonetheless there is still interesting information in the paper. the fact that NBEAL2 appears to be absent from Treg is a surprise and may explain some aspects of the data, but equally it makes it hard to understand the relationship between LRBA and NBEAL2 in terms of redundancy / essentiality for CTLA-4 control.

We thank the reviewer for highlighting the interesting information in our paper.

As noted by reviewer #3, we indeed used different antibody panels for the Tregs and activated T cells labeling. However, our goal was to compare the expression of different markers, and in particular of CTLA-4, between the Tregs from patients and healthy controls on the one hand, and between the activated T cells from patients and healthy controls on the other hand. It would be very risky to compare the expression of CTLA-4 on Tregs (from thawed PBMCs) with the expression on conventional T cells, which have been activated *in vitro* for several days. This is another reason why we never intended to compare the expression of CTLA-4 between these cell subtypes. Lastly, as stated in the methods, healthy donors' cells were stained, acquired, and analyzed on the same day with the same settings as patients' cells. To clarify these comparisons, the sentence (line 245) has been modified.

We agree with reviewer#3 that our results raise new questions about the precise role of NBEAL2 and LRBA in conventional T cells. Indeed, it seems that they are not redundant in these cells since the deficiency of one or the other leads to a lack of expression of CTLA-4, as shown by our experiments using CRISPR/Cas9 carried out in activated T cells of healthy controls (which is recapitulating the observation made in the patients' cells). It remains possible that these molecules act together to lead to membrane re-expression of CTLA-4 or that they act at different stages of this cellular process. However, overall, our results show for the first time that NBEAL2 deficiency is associated with CTLA-4 deficiency in conventional T cells. This might eventually leads to targeted therapy in GPS patients who develop autoimmune manifestations.

Reviewer #4 (Remarks to the Author):

All my comments have been addressed.